# Inefficient prioritization of task-relevant attributes during instrumental information demand

Isabella Rischall[1,2,5], Laura Hunter[1,2,5], Greg Jensen[1,2,3] & Jacqueline Gottlieb [1,2,4] ✉

In natural settings, people evaluate complex multi-attribute situations and decide which attribute to request information about. Little is known about how people make this selection and specifically, how they identify individual observations that best predict the value of a multi-attribute situation. Here show that, in a simple task of information demand, participants inefficiently query attributes that have high individual value but are relatively uninformative about a total payoff. This inefficiency is robust in two instrumental conditions in which gathering less informative observations leads to significantly lower rewards. Across individuals, variations in the sensitivity to informativeness is associated with personality metrics, showing negative associations with extraversion and thrill seeking and positive associations with stress tolerance and need for cognition. Thus, people select informative queries using sub-optimal strategies that are associated with personality traits and influence consequential choices.

In many conditions, people must decide between alternative propositions or courses of action and, perhaps more primarily, must select which information to gather for guiding their actions. Choices of information have been de-emphasized in traditional research but are increasingly investigated in recent literature on information demand that examines how people become motivated to request or avoid information (see refs. 1,2 for recent reviews). These studies have used diverse methodologies, including probing how participants request information about simple simulated scenarios involving personal finance or health[1,3,4] or how they choose among "lotteries" that provide probabilistic rewards with different value, uncertainty, and information availability[2,5–10].

Together, the studies suggest that information demand is driven by several factors[1,2]. On one hand, people seek to obtain information that guides utility-relevant actions, suggesting that they are motivated by the instrumental rewards they obtain when acting on the information. On the other hand, people also seek to obtain information that cannot be exploited for reward gains (is not instrumental), suggesting

that they value information as a good in itself. Moreover, the value of non-instrumental information reflects two interacting motives—a desire to reduce uncertainty as early as possible and a drive to obtain observations that predict favorable but not unfavorable outcomes (e.g., a reward rather than a lack of reward)[6,7].

The latter, value-based, motive is striking because it can oppose uncertainty-dependent information gathering and drive people to *avoid* information and choose ignorance (higher uncertainty) if the outcome is likely to be unfavorable[1,2,4,6,7,9,11,12]. The common explanation of these value effects is in terms of anticipatory utility—a hedonic or emotional bias that motivates people to anticipate and savor desirable outcomes but avoid the anticipation and dread of undesirable outcomes (e.g. refs. 1,12). However, key open questions remain about the nature and scope of value-based biases in information gathering.

One significant question is how value effects differ across choice situations. Many studies of information demand test how participants trade off advance information against the rewards they obtain, by

[1]Department of Neuroscience, Columbia University, New York, NY, USA. [2]Mortimer B. Zuckerman Mind Brain Behavior Institute, Columbia University, New York, NY, USA. [3]Department of Psychology, Reed College, Portland, OR, USA. [4]Kavli Institute for Brain Science, Columbia University, New York, NY, USA. [5]These authors contributed equally: Isabella Rischall, Laura Hunter. ✉e-mail: jg2141@columbia.edu

asking them to choose among alternative lotteries that vary independently in their expected value (EV), uncertainty, and information availability. In many natural settings, however, people choose among multiple observations that are relevant to a single eventual outcome. For example, in question-asking scenarios, participants are faced with a task (e.g., solving a puzzle or making a categorization decision) and can choose between questions (queries) that are relevant to the task[13–16]. Translated to a lottery task, this implies a distinct choice situation, in which participants receive combined rewards from several lotteries and choose which lottery to inquire about to best predict the total eventual outcome. Thus, rather than choosing among competing sources of income, one chooses among competing observations that are relevant to a single eventual income.

Studies of question-asking abilities show that in such situations participants often generate inefficient, suboptimal queries[13–16], but the nature of these inefficiencies are not well understood. A specific question arising from the discussion above is whether choices among alternative observations are subject to value confounds.

This question is brought into focus by normative theories, which predict that observations should be prioritized strictly based on their ability to resolve uncertainty about the full situation regardless of any individual value that may be associated with the observation[7,17]. To illustrate this prediction, imagine that you want to predict the total cost of a vacation that includes a hotel and a car. The two items may independently differ in value and prior uncertainty; e.g., the hotel cost may be higher but more predictable than that of the car. If you only have time to inquire about the precise cost of one of the items, which item should you prioritize to best predict the total cost of the package? The normative strategy is to query the item that has the highest prior uncertainty (is most difficult to predict in advance) as this will most efficiently reduce your uncertainty regarding the total. The value of the individual item, however (e.g., whether the hotel costs more than the car) should be irrelevant to your inquiry as it does not affect the uncertainty that the observation resolves.

Contrary to this clear prediction, a recent report from our laboratory suggests that value-based biases affect the demand for alternative observations[7]. Kobayashi et al. used a task similar to the scenario above, in which participants received combined payoffs from two lotteries but could only request advance information about the precise prize from one lottery. They found that a vast majority of participants inquired about the lottery with the higher individual value even when the inquiry resolved less uncertainty about the total payoff. Importantly, the findings could not be explained by a reinforcement learning model of anticipatory utility in which participants are assumed to assign value to observations recursively, based on the total rewards an observation predicts to accrue later on[7,12]. Instead, the findings imply that people myopically savor individual observations regardless of the total future payoffs. That is, they value the opportunity to learn about a high-value individual prize regardless of the uncertainty this resolves about the full situation.

However, two features of the study of Kobayashi et al. limit its broader applicability. First, the study focused on non-instrumental conditions in which participants had no monetary incentives to minimize their uncertainty, raising the possibility that people adopted arbitrary strategies that may vanish in more important instrumental conditions. Second, while the study showed that information demand had individual variability, we found no evidence that this variability was associated with a limited set of personality scores, which contrasts with recent reports that did find such associations in lottery-based tasks[10] and simulated scenarios involving personally relevant information[3,4]. Since personality metrics are stable over multiple tasks, their association with information demand may provide valuable bridges between studies in multiple settings.

In the present experiment, we examined both questions by extending the task of Kobayashi et al. to include two distinct instrumental conditions and examining a broader range of personality metrics. We show that inefficient information gathering is replicated in instrumental conditions despite leading to much lower rewards. Moreover, inter-individual variability in sampling efficiency was associated with personality traits, in particular need for cognition, extraversion, and the tolerance for uncertainty.

## Results
### Task
Participants ($n = 610$) performed an online task in which they attempted to predict the sum of two random draws by inquiring about the precise value of one of the draws. Participants were shown two lotteries, each of which could deliver one of two possible point quantities indicated with numbers and markings on a 500-point scale (Fig. 1a). One lottery had a high variance (hiVar, 120 point difference between the possible values) and the other had a low variance (loVar, 30 point difference). The relative EV of the lotteries varied independently of uncertainty so that, across trials, the hiVar lottery could have lower, equal, or higher EV relative to the loVar lottery. Participants were told that one point quantity would be drawn from each lottery, randomly and with equal probability, and the trial's payoff would be the sum of the draws. The precise values of the draws were not shown by default but participants were asked to inquire about one of the draws to estimate the total payoff.

We presented the same information sampling stage in three tasks that differed in the post-sampling steps—the actions that participants took after gathering information (Fig. 1b). In the Observe task, the information was non-instrumental, as there were no actions that people could take to alter the draws, whereas, in the Estimate and Intervene tasks, participants made instrumental decisions based on the information they sampled. In the Estimate task, after revealing a draw, participants were prompted to guess if the sum of the draws was above or below a criterion of 500 points (the sum of the lottery EVs; Fig. 1b, middle). If the guess was correct, the payoff was equal to the sum of the draws but, if the guess was incorrect, the payoff was 0 points. Thus, participants were incentivized to request the information they believed would best allow them to predict the total payoff. In the Intervene task, participants could improve the draw from the lottery they inspected. After inquiring about one lottery, participants could decide if they wanted to keep the draw they obtained or switch to the average EV of the lottery (Fig. 1b, right). The payoff was equal to the draw from the unrevealed lottery (which was hidden and beyond the participants' control) plus the draw from the inspected lottery (which was known and potentially altered). Thus, participants were incentivized to inspect the lottery from which they could recuperate the largest amount if they happened to obtain a low draw.

Each task was presented in blocks of 126 trials and was preceded by detailed instructions, practice trials, and quiz questions to ensure comprehension (see the "Methods" section). Participants received no feedback about the sum of the draws or the accuracy of their instrumental decision. At the end of each block, one trial was randomly chosen from those the participants had played, and its payoff determined the bonus (at a conversion rate of 2000 points = $1, announced in advance). To test for possible strategy transfer between instrumental and non-instrumental conditions, we presented the Observe task first, followed by the Estimate and Intervene tasks counterbalanced in the 2nd and 3rd blocks and a final block repeating the Observe task.

### Reasoning about information gathering
In each task, the optimal strategy for minimizing uncertainty regarding the sum was to reveal the prize from the hiVar lottery regardless of the EV of this lottery. The task was sufficiently simple that this strategy could be deduced from the description alone without the need for learning or elaborate computations. To illustrate this reasoning, Fig. 1c

traces the outcomes that were possible in a representative trial after inspecting each lottery. If the participants inspected the loVar lottery and observed a high draw, they could expect to receive that draw (315 points) plus an uncertain amount of either 260 or 140 points from the unobserved lottery (Fig. 1c, top row). Thus, the possible total draws were 575 and 455 points and their range was 120 points, equal to the range of the unobserved hiVar lottery. Alternatively, if participants inspected the hiVar lottery and observed the high draw, they could expect to receive 260 points plus either 315 or 285 points from the loVar lottery (Fig. 1c, bottom row). The possible sums were 575 and 545 points and their range was only 30 points− equal to the range of the loVar lottery. Thus in both cases, the uncertainty of the sum was equal to the uncertainty of the unobserved lottery, and was therefore minimized after inspecting the hiVar lottery. Note also that observing a high-value individual draw did not necessarily predict a higher total payoff. In the examples above, the highest possible total value was 575 points for both observing decisions and the lowest possible total value was lower after the higher-value observation than after the low-value observation (455 vs. 545 points, top vs bottom panels). Thus, the EV of the individual observed lottery was irrelevant to the total payoff. Only the uncertainty of this lottery affected the uncertainty about the payoff.

In instrumental conditions, a less uncertain estimate of the total translated into higher instrumental rewards. In the Estimate task, after revealing a high or low draw from the hiVar lottery, participants could be certain that both possible sums were, respectively, higher or lower than the criterion of 500 points (Fig. 1c, bottom). In contrast, an observation from the loVar lottery was equally consistent with the sum being higher or lower than the criterion. Thus, revealing the hiVar versus the loVar lottery was associated with a doubling of the expected accuracy of the guess from 50% to 100% (Fig. 1c, top). In the Intervene task, if participants observed a low draw, they could exchange it for the average of the inspected lottery. Because the respective differences between a low draw and the mean were 60 vs. 15 points, participants could recuperate 4 times more points by inspecting the hiVar versus the loVar lottery. Supplementary Tables 1 and 2 and the related discussion show further detail on these calculations, as well as the earnings that participants missed due to their sampling. Note, however, that precise calculations were not necessary for good performance; qualitative reasoning sufficed to determine that instrumental rewards could be 2−4 times higher after revealing the hiVar relative to the loVar lottery.

## Inefficient sampling persists in instrumental conditions

To analyze the participants' sampling, we plotted the average probability of revealing the hiVar lottery (%reveal hiVar) as a function of the relative EV of this lottery (ΔEV, defined as the EV of the hiVar lottery minus the EV of the loVar lottery). The uncertainty-minimizing strategy was a function that was flat at 100%, indicating sampling that depended strictly on uncertainty and was insensitive to EV (Fig. 2a, black). The participants' choice functions were shifted upward toward values greater than 50%, suggesting a bias toward sampling the hiVar lottery, but this bias was far from the normative strategy, and importantly, the functions had strong positive slopes showing a preference for the

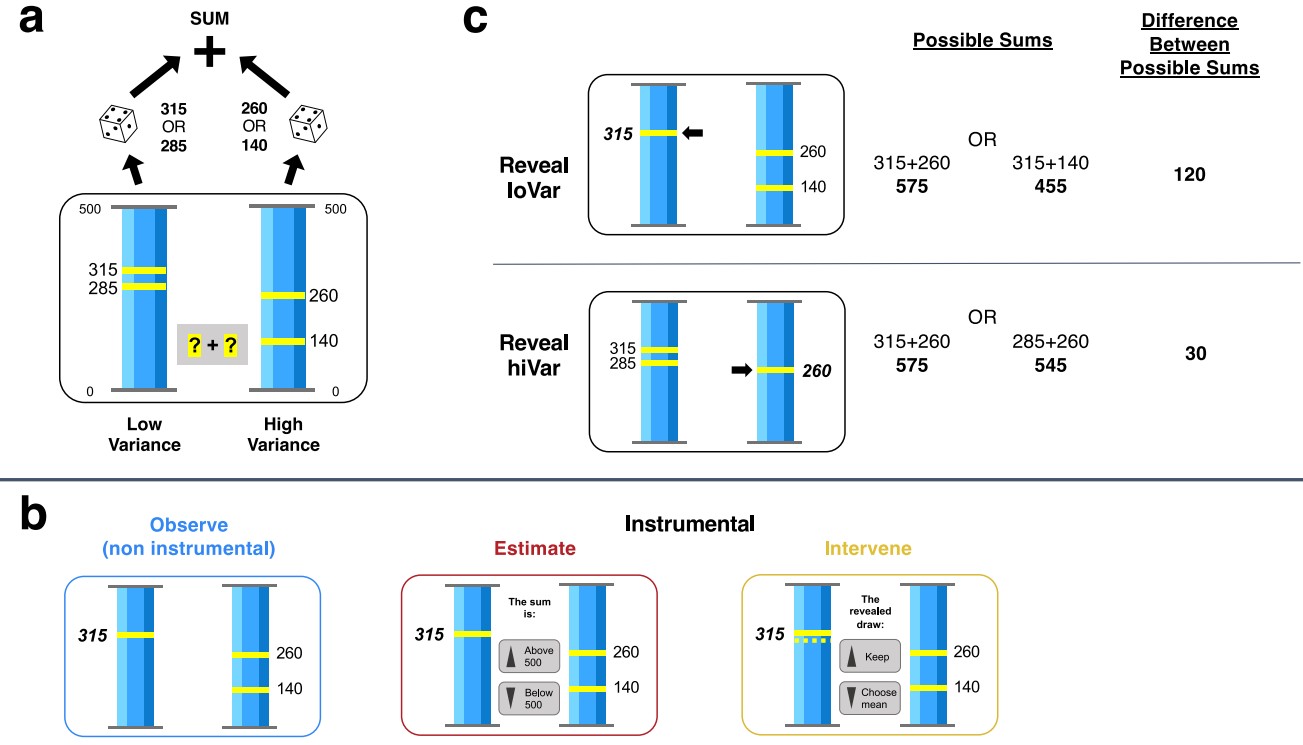

**Fig. 1 | Task. a** Lotteries and generation of payoffs. On each trial, participants received two "lotteries", each comprising two point prizes depicted with numbers and vertical markings (yellow lines) on a vertical scale (blue column). The two lotteries had, respectively, low or high variance and different relative EV. To generate the payoff, the computer randomly drew one prize from each lottery and calculated the sum of the prizes. The precise prizes that were realized were kept secret ("?") but participants requested to reveal the precise prize from one lottery. The sampling step was identical across all task blocks. **b** Post-sampling actions. Participants completed three variants of the task that differed in the actions they were required to perform after gathering information. In the Observe (non-instrumental) block, participants merely progressed to the next trial after revealing a prize (in this example, 315 points). In the Estimate block, participants made a second decision about whether the sum of the prizes was greater or smaller than 500 points. In the 'Intervene' block, participants made a second decision whether to keep the prize they revealed or exchange it for the average (EV) of the lottery. In all figures, we use blue for Observe, red for Estimate, and yellow for Intervene blocks. **c** Prospective reasoning about information gains. Reasoning about the possible sums upon the reveal of the high draw from the loVar (top) or hiVar (bottom) lottery in the example trial in (**a**). See text for further details.

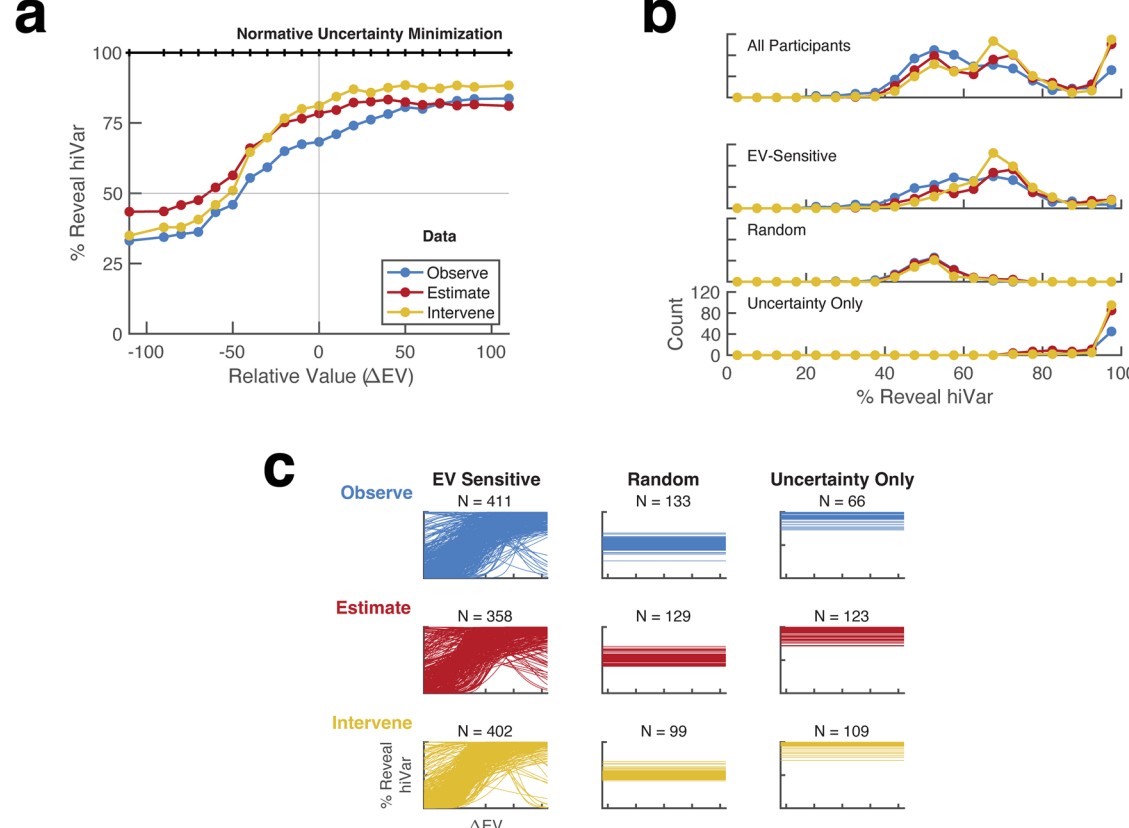

**Fig. 2 | Sampling behavior. a** Average psychometric curves showing the percentage of trials in which the participants revealed the high-variance lottery as a function of the difference between the lottery EVs (ΔEV, defined as the difference between the EV of the hiVar minus that of the loVar lottery). The colored curves show the average behavior across all participants (*n* = 610; blue: Observe, red: Estimate; yellow: Intervene). Error bars are omitted for clarity. **b** Model-free

measure of sampling efficiency and model-based categorization. The traces show the distributions of %reveal hiVar by task (color-coded as in **a**), for all participants (top) and for participants classified according to the modeling analysis (bottom). **c** Best fitting psychometric curves for each participant, categorized as indicating an EV-sensitive, Random, or Uncertainty-only strategy. *N* indicates the number of participants in each category.

higher-EV lottery (Fig. 2a, colored traces). This was the case even in instrumental conditions when this preference led to the loss of rewards.

## Sampling shows individual variability

Consistent with the findings of Kobayashi et al., individual participants combined sensitivity to both uncertainty and EV. To parametrize this dual sensitivity and its individual variability, we first fit each participant's data with a sigmoid function in which the intercept (a) captured sensitivity to uncertainty and the slope, (b), captured sensitivity to EV (see the "Methods" section, Eq. (1)). Preliminary analyses showed that, while this strategy worked well in most cases, a sizeable minority of participants who had high uncertainty sensitivity were fit with strongly negative intercepts due to minor variations in their (non-significant) slopes (Fig. S1a; *b* ~ 0). To circumvent this ambiguity, we explored an alternative approach in which we used model comparisons to determine, for each participant, if their choices were better fit with the two-parameter sigmoid function above, versus a one-parameter function characterized only by an intercept *c* (see the "Methods" section, Eq. (2)). This reduced the indeterminacy of the fits and produced a monotonic relationship between %reveal hiVar and the intercept parameters (*a* or *c*), while capturing EV sensitivity through the separate parameter *b* (Fig. S1a vs. S1b).

This procedure, in turn, suggested that participants could be classified as being sensitive or insensitive to EV based on whether their data were better fit by the sigmoid or flat functions. To evaluate the validity of this view, we compared the model-based classification with

the overall probability of revealing the hiVar lottery (%reveal hiVar), which is proportional to earnings in instrumental conditions (Supplementary Table 1) and serves as a simple model-free measure of sampling efficiency.

The analysis rejected the hypothesis that the distributions of % reveal hiVar values were unimodal (Hartigan's Dip Test, *n* = 610 participants, bootstrapped *p* < 0.001 in each task) and suggested that participants adopted three distinct strategies that mapped onto the model-based classification (Fig. 2b top). The first and main mode of the distributions included participants whom we designated as EV-sensitive based on the lower BIC values when fit with a sigmoid relative to a flat function (Fig. 2b, second row; Fig. S1a). All participants in this group showed significant slope parameters (*b*; 95% confidence interval (CI) did not include 0), and the vast majority showed positive slopes indicating a preference to inspect the high-EV lottery (Fig. 2c, left; *b* was positive (rather than negative) for 97%, 93%, and 97% of EV-sensitive participants in, respectively, Observe, Estimate, and Intervene blocks). Further confirming their EV sensitivity, these participants showed reaction times for the sampling decision that peaked at low values of ΔEV (Fig. S2). Finally, individuals in this group had %reveal hiVar ranging between ~50% and ~80%, falling on a continuum of reward and uncertainty sensitivity as reported by Kobayashi et al.[7] (Fig. S1a and further below).

The choices of the remaining participants were classified as insensitive to EV based on their lower BIC when fit with the flat rather than sigmoid function and, in turn, fell into two distinct groups. One group had parameter *c* that was not significantly different from

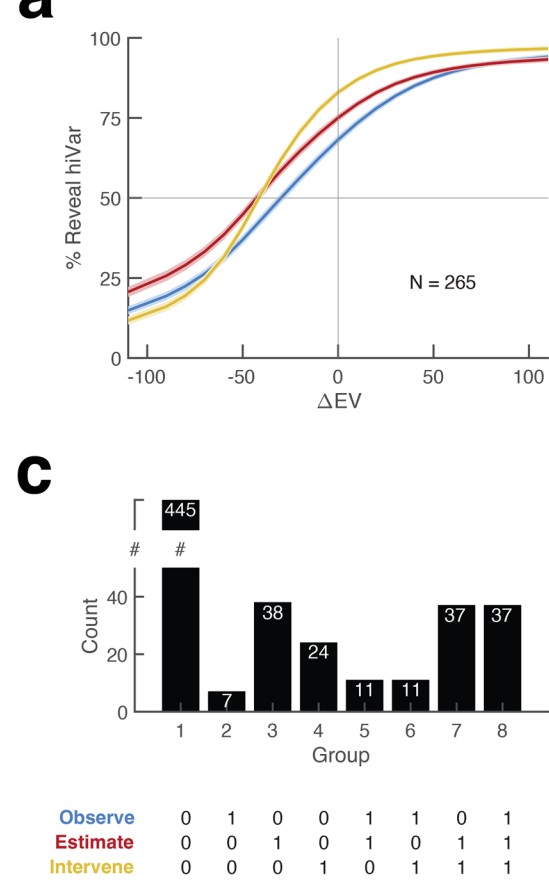

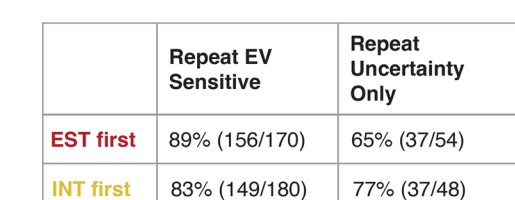

**Fig. 3 | Consistency across blocks. a** Best fit psychometric functions averaged across the $n = 265$ participants who showed EV-sensitive strategies in all 3 blocks. The blue, red, and yellow curves correspond to data from Observe, Estimate, and Intervene respectively. The shaded areas correspond to the standard area of each curve. **b** Paired comparisons between the Estimate and Intervene relative to the Observe block, for the 256 participants in (**a**). The panels show the distributions of the differences between the best-fit intercepts (*a*), slopes (*b*), and raw percentages of revealing the hiVar lottery in the instrumental task and the first Observe block (top row, red: Estimate task; bottom row, yellow: Intervene task). The white lines show medians and stars denote $p < 0.001$ for each distribution relative to 0 (Wilcoxon signed-rank test). In the middle panel, the outer stars/vertical line shows

$p < 0.001$ for Estimate vs. Intervene, Wilcoxon signed-rank test. **c** Strategies across blocks. All participants ($n = 610$) were categorized into 8 groups based on the combination of blocks in which they showed uncertainty-only strategies (1 in *x*-axis label). The white numerals show the number in each group (note the break in the vertical scale). Only groups 7 and 8 showed efficient sampling in both instrumental blocks. **d** Strategy changes across the two instrumental blocks. Participants were separated by the order of performing the Estimate and Intervene tasks and counted according to whether, given that they started with an EV-sensitive or Uncertainty-only strategy in the first instrumental block, they repeated the same strategy in the second instrumental block.

chance (95% CI included 0.5) indicating that their sampling did not significantly depend on uncertainty or EV (Fig. 2c, Random). This participant group had overall low accuracy for the post-sampling instrumental decisions (even in the Intervene task in which this decision was trivial; Supplementary Table 2) suggesting that they were inattentive or disengaged from the task. Thus, although the % reveal hiVar values of Random participants overlapped with those of the EV-sensitive group (Fig. 2b), these participants pursued a distinct strategy.

Importantly, the remaining EV-insensitive participants had parameter *c* that was significantly >0.5 (95% CI was above 0.5) indicating a near-optimal strategy focused on the hiVar lottery (Fig. 2c, Uncertainty-only). This group formed a distinct mode of the %reveal hiVar distribution with values > 90% (Fig. 2b, bottom) and showed reaction times that were insensitive to ΔEV (Fig. S2), confirming their model-based designation. We verified that the classification results were robust to the choice of criterion (Fig. S1b), showing that our parametrization reliably quantified individual sensitivity to uncertainty and EV and identified meaningful information-gathering strategies.

## Instrumental demands modestly enhance uncertainty sensitivity

We next used parametrization above to compare results across tasks. The findings suggested that sampling efficiency was enhanced by instrumental incentives, but the increases were modest and included changes in both uncertainty and EV sensitivity.

We first focused on a subset of 265 participants who pursued EV-sensitive strategies in all tasks (Fig. 3a, b). Comparisons of the choice functions suggested that these participants had higher uncertainty sensitivity (an upward shift of the functions) in the Intervene and Estimate relative to the Observe task (Fig. 3a). This was confirmed by the distributions of differences between parameter *a*, which were significantly greater than 0 in comparisons between Intervene and Observe (Fig. 3b, left, mean ± standard error (SEM) of the differences: $0.38 \pm 0.02$; two-tailed Wilcoxon signed-rank test vs. 0, $n = 265$ participants) and between Estimate and Observe ($0.39 \pm 0.02$; $p < 0.001$ vs. 0, two-tailed Wilcoxon signed-rank test vs. 0, $n = 265$ participants). This result was not explained by the order of presentation of the Estimate and Intervene tasks (2-way ANOVA on the parameter *a*; effect of the order, $p = 0.757$, $F(\text{df} = 794) = 0.10$, effect size $\eta^2 = 0.0014$;

interaction between order and instrumentality, $p = 0.300$, $F(df = 794)$ $= 1.08$, effect size $\eta^2 = 0.20$, $n = 265$ participants). Finally, as documented in detail in Fig. S3a,b, an increase in uncertainty sensitivity in instrumental conditions was replicated in participants who pursued EV- sensitive strategies only in some of the tasks and in participants pursuing Uncertainty-only strategies.

Although the increases in uncertainty sensitivity were reliable and consistent, they resulted in only modest improvements in sampling efficiency. The EV-sensitive group showed median increases in %reveal hiVar of, respectively, 4% and 6.4% in the Estimate and Intervene relative to the Observe task (Fig. 3b, right). To more specifically measure how efficiency was impacted by these samplers' uncertainty and EV-sensitivity, we reasoned that their %reveal hiVar values at $\Delta EV = 0$ measured the contribution of uncertainty sensitivit, while the difference between these values and overall %reveal hiVar (across all levels of $\Delta EV$) measured the additional contribution of EV sensitivity. In the Observe, Estimate, and Intervene tasks %reveal hiVar at $\Delta EV = 0$ was below optimal by, respectively, $31.79 \pm 1.15\%$, $24.96 \pm 1.17\%$, and $17.00 \pm 1.03\%$. The overall %reveal hiVar values were an additional $6.46 \pm 1.31\%$, $8.53 \pm 1.16\%$, and $15.15 \pm 0.97\%$ lower than the values at $\Delta EV = 0$. Thus, participants had losses of efficiency of 17–32% due to imperfect sensitivity to uncertainty and additional losses of 6–15% due to sensitivity to $\Delta EV$.

### Sampling strategies are inconsistent across tasks

A possible explanation for the inefficient sampling we find is that participants failed to identify informative observations based on the reasoning in Fig. 1b. Two aspects of our findings support this interpretation.

First, had participants identified the optimal strategy, we would expect them to sample identically in the two instrumental conditions in which this strategy was incentivized. Contrary to this view, sampling showed significant differences between the Intervene and Estimate tasks. Participants who consistently pursued EV-sensitive strategies had higher value sensitivity in the Intervene relative to the Estimate task (Fig. 3b, middle panel; paired differences in parameter $b$, $1.70 \pm 0.232$; $p < 0.001$ two-tailed Wilcoxon signed-rank test vs 0, $n = 265$ participants) as also found in those who pursued EV-sensitive strategies in some of the tasks (detailed statistics in Fig. S3a, b). We found no credible evidence of correlations between the task effects on parameters $a$ and $b$ (task difference between the Intervene and Estimate task in parameter $a$ versus the difference in parameter $b$, Spearman's $r = 0.06$, $p = 0.319$, $n = 256$ participants). Uncertainty-only participants had higher uncertainty sensitivity in the Intervene vs the Estimate task (detailed statistics in Fig. S3a, b). Thus, although the optimal sampling strategy was identical in all tasks, participants made task-specific adjustments in their sampling strategy including independent adjustments in uncertainty and EV sensitivity.

Second, we reasoned that, if participants had correctly inferred the optimal strategy in the first instrumental block they encountered, they should repeat it in the second instrumental block. To evaluate this possibility, we measured the probabilities that a participant would repeat an Uncertainty-only strategy in the second instrumental block, conditional on having adopted that strategy in the first block. Conditioning on the initial strategy controlled for base rates and allowed us to compare this probability against that of repeating an EV-sensitive strategy. We found no credible evidence that participants were more likely to repeat an Uncertainty-only relative to an EV-sensitive strategy (Fig. 3d). In fact, a significant difference in the opposite direction was found in the full data set (Fig. 3d; one-way chi-square test, $\chi^2$ (df = 1) = 12.42, $p < 0.001$, effect size $V = 0.17$, $n = 452$) and in the participants who encountered the Estimate task first ($\chi^2$ (df = 1) = 18.57, $p < 0.001$, effect size $V = 0.41$, one-way chi-square test, $n = 224$) with no credible evidence for a difference in those who encountered *Intervene* first ($\chi^2$ (df = 1) =

0.82, $p = 0.366$, effect size $V = 0.09$, one-way chi-square test, $n = 228$). Consistent with this finding, only 11% of participants pursued Uncertainty-only strategies in both the Intervene and Estimate tasks while 14% inconsistently switched strategies across tasks (Fig. 3c; groups 7–8 vs. groups 3–6; see Fig. S4 for the full choice functions across tasks). Thus, some participants who used an Uncertainty-only strategy appeared not to have realized that this strategy was optimal in all tasks.

### No credible evidence that sampling the loVar lottery was related to the uncertainty of the post-sampling decision

A possible explanation for the participants' sampling of the loVar lottery is that they believed, incorrectly, that the prize they observed in this lottery was correlated with the prize they would obtain from the hiVar lottery and, thus, was predictive of the total payoff. If this hypothesis were correct, we expect that, in the Estimate task, participants would be equally confident when making a guess based on a loVar and hiVar observation. Contrary to this view, reaction times for the post-sampling decision (i.e., the time participants took for estimating the sum) were much longer after observations from the loVar versus the hiVar lottery, suggesting that participants had lower confidence in the former case (Fig. S5a; $p < 0.001$; Wilcoxon signed rank test, $n = 358$ participants with EV sensitive strategies in the Estimate task). Moreover, the probability that a participant inspected the loVar lottery was negatively correlated with the probability that they "obeyed" the observation from this lottery (i.e., estimated the sum to be high or low when the revealed prize was correspondingly high or low; Fig. S5b; Spearman's $r = -0.527$, $p < 0.000$; $n = 358$ participants). Thus, we found no credible evidence that participants overestimated the informativeness of the loVar lottery.

### Learning and trial-by-trial adjustments

Although we attempted to minimize learning by withholding trial-by-trial reward feedback, participants may have learned based on other clues in the task. One potential mechanism is that participants would increase or decrease the subjective value they ascribe to a lottery based on whether the lottery signaled a high or low prize in the previous trial. However, a sequential trial analysis provided no credible evidence for this view, as observing a high (low) draw on one trial did not alter sampling on the following trial (Fig. S6a) nor was the magnitude of the prior-trial effect correlated with individual EV sensitivity (Fig. S6b).

An alternative possibility is that participants improved their sampling simply by spending more time on the task and reflecting on the optimal strategy. However, as noted in Fig. 3d, we found no credible evidence that sampling efficiency improved between the first and second instrumental blocks. Similarly, we found no credible evidence that efficiency in non-instrumental conditions improved after performing the instrumental conditions—i.e., in the second relative to the first Observe blocks (Fig. S7).

Our findings, however, support a third possibility—that some participants adjusted their sampling by monitoring their instrumental decisions. Participants with non-random strategies showed modest but significant increases in sampling efficiency during the course of a block in instrumental but not non-instrumental conditions (Fig. S6c). Importantly, these improvements were strongest in participants with Uncertainty-only strategies who had already high levels of hiVar sampling at the start of the block. At both the group and individual levels, Uncertainty-only participants showed significant increases in sampling efficiency in both the Intervene and Estimate tasks, while EV-sensitive participants did so only in the Estimate task (see Fig. S6c for details). Thus, participants seem to have used clues from the instrumental decision in ways that were gated by their initial understanding of the optimal strategy.

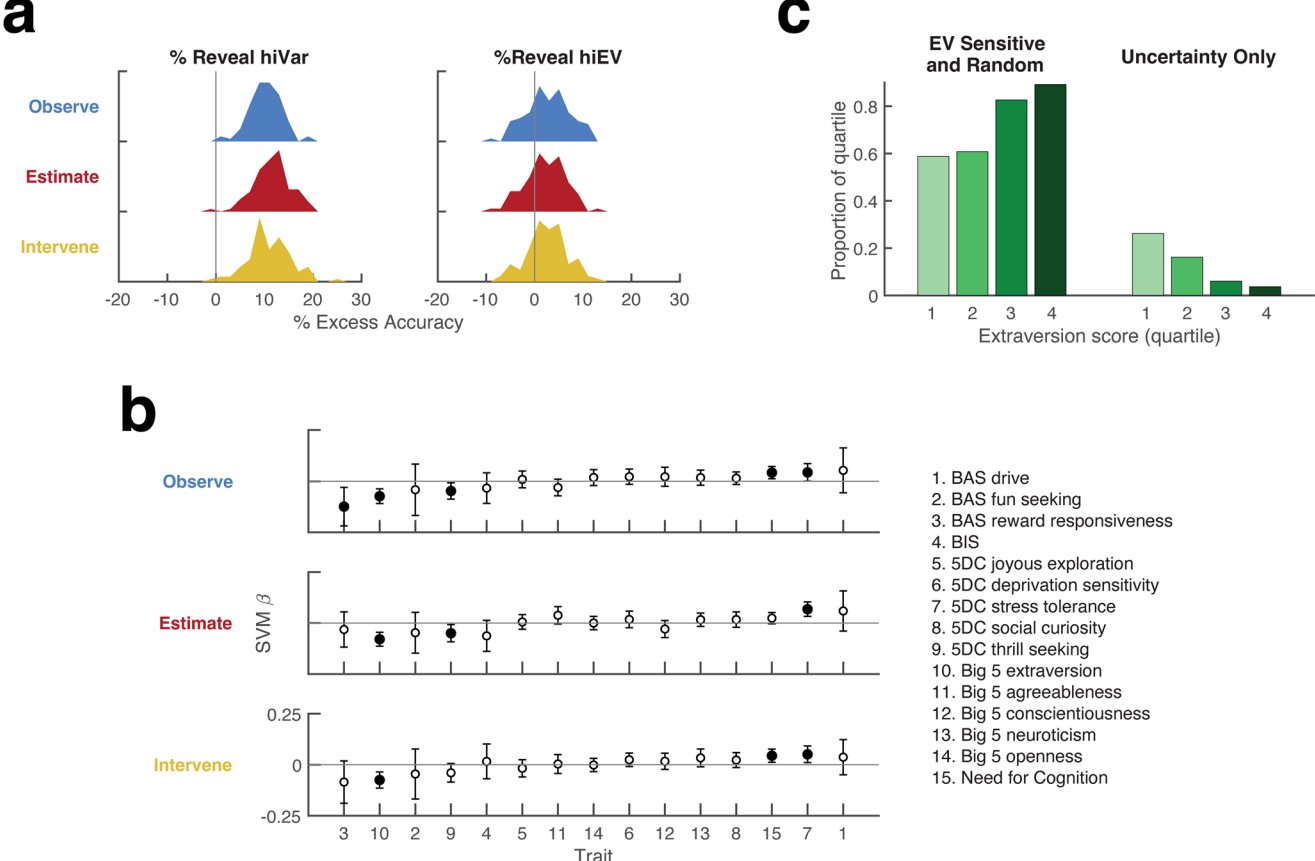

**Fig. 4 | Associations with personality scores. a** Decoding of uncertainty-based sampling based on personality and demographic indicators. The distribution of SVM decoding accuracy for classifying participants into whether their %reveal hiVar or %reveal high EV was above vs. below the median. Excess accuracy is the percentage of correct classifications when the SVM was trained on true data minus the percentage of correct classifications when the SVM was trained on randomized data. Distributions are over 100 bootstrap iterations with 50:50 cross-validation. **b** Trait coefficients (β) from the SVM classification boundary ordered by the average magnitude of the coefficients across all tasks. The points show the mean and 95% confidence intervals over 100 bootstrap iterations; black indicates $p < 0.05$ relative to 0. Demographic indicators (age, sex, and education) were included as predictors in each SVM but did not show significant coefficients and are omitted from the figure for clarity. **c** Instrumental sampling as a function of extraversion. Participants were divided into quartiles based on extraversion scores (green shading) and into two groups based on their sampling strategies (EV-sensitive or random in both instrumental blocks (groups 1–2 in **c**) or Uncertainty-only in both instrumental blocks (groups 7 or 8 in Fig. **c**). The y-axis shows the percentage of each extraversion quartile falling in each strategy group.

## Sampling strategies relate to personality data

To determine if information demand relates to personality traits, we obtained ratings on 5 personality questionnaires, including the Behavioral Inhibition-Approach Scale[18] (BIS/BAS), Big 5 Personality Inventory[19], Need for Cognition[20], and 5-Dimension Curiosity Scale[21]. A multiple-regression analysis (see the "Methods" section) produced no credible evidence that value and uncertainty sensitivity were reducible to personality constructs, as individual personality scores typically explained much less than 1% of the variance in the sensitivity to uncertainty or EV with no score explaining more than 2.5%. Nevertheless, a support-vector machine (SVM) trained using all the personality metrics produced above-chance classification of sampling efficiency, showing that our constellation of personality metrics captured significant features of information gathering.

Training SVM classifiers to predict sampling efficiency (high/low % reveal hiVar classification based on a median split; see the "Methods" section) produced classification accuracy that was reliably above chance in all tasks (Fig. 4a, left; across 100 cross-validated iterations, means and 95% confidence intervals (CI) for excess accuracy (accuracy minus chance) were 10.2% [9.5%, 10.8%] in the Observe task, 11.6% [11.0%, 12.4%] in the Estimate task and 11.1% [10.3%, 12.0%] in the Intervene task). In contrast, training the classifiers to predict EV sensitivity produced much lower accuracy (Fig. 4a, right; Observe, 2.7%

[1.8%, 3.6%]; Estimate, 2.1% [1.3%, 3.0%]; Intervene, 2.6% [1.8%, 3.4%]; see the "Methods" section). Moreover, we found no credible evidence of consistent associations between EV sensitivity and individual personality scores, reinforcing the view that uncertainty and reward sensitivity are distinct constructs.

Further analyses identified specific individual scores that were consistent predictors of sensitivity. The most consistent predictor in all tasks was the Big5 extraversion score (Fig. 4b). This score had a negative weight, indicating that more efficient sampling was associated with low extraversion. This was confirmed by a second analysis of participants who adopted Uncertainty-only strategies in both instrumental blocks, showing that, in contrast with the other strategy groups, these participants fell primarily in low-extraversion quartiles (Fig. 4c).

An additional predictor of higher efficiency sampling was a higher need for cognition, consistent with the idea that identifying informative observations required reflection and inference about the task structure (Fig. 4c). Finally, %reveal hiVar was predicted by two uncertainty subscales of the 5DC curiosity questionnaire, including thrill seeking with negative weights and stress tolerance with positive weights (Fig. 4c). As we elaborate in the "Discussion" section, these findings suggest an intricate association between uncertainty sensitivity as measured by our task and the curiosity scale. All the results

were replicated using logistic classification and linear regression analyses, confirming the reliable association between these personality metrics and uncertainty sensitivity in our tasks.

## Discussion

We investigated information demand in a question-asking scenario in which people prioritized information about different lotteries that contributed to a single payoff. We show that a large majority of participants generated inquiries that failed to minimize the uncertainty about the total payoff. The results replicated in three distinct task versions, showing that inefficient prioritization persists in instrumental conditions when it is associated with two- or four-fold lower reward probability. Finally, we show that sampling efficiency was associated with personality traits, helping link studies of information gathering across different domains of research.

Our analytical method for measuring the influences of uncertainty and EV was similar to that of Kobayashi et al., but added a classification of strategies as falling into EV-sensitive, Random and Uncertainty-only categories. Several observations suggest that this classification captured meaningful aspects of the behavioral data. First, we confirmed the conclusions of Kobayashi et al. that the EV-sensitive group—the largest class in our study—showed reward and uncertainty influences that were captured by independent parameters and showed marked inter-individual variability that fell along a continuum of uncertainty and reward sensitivities. Second, our classification procedure identified a distinct group of participants who sampled randomly and were likely to have been disengaged from the task. Finally, and particularly important, our procedure correctly identified a small subset of participants who formed a distinct mode of the %reveal hiVar distribution and pursued strictly uncertainty-based strategies. Some of the participants in this group seem to have correctly recognized the normative, EV-insensitive strategy, as they had Uncertainty-only strategies in all tasks (or at least in the two instrumental conditions). However, others used EV-sensitive strategies in some of the tasks, suggesting that they lacked a true understanding of the optimal strategy. In sum, our results confirm those of Kobayashi et al. and refine the analysis of the participants' strategies.

As we note in the "Introduction" section, the sensitivity to EV we describe may have a distinct computational substrate than that of the phenomenon of anticipatory utility[7]. A reinforcement learning model of information demand proposed that anticipatory utility arises when informational states gain positive or negative value by virtue of producing, respectively, positive or negative reward prediction errors (RPEs)[12]. However, as explained in detail in Kobayashi et al.[7], the model assumes that participants assign value to a state by recursively aggregating all the outcomes that are expected to follow that state. Thus, the model predicts that anticipatory utility depends on the total EV that follows information gathering and cannot explain why participants favored a lottery with higher individual EV, which was independent of the total (constant) EV.

Our results thus have two possible explanations. One possibility is that participants showed preferences over information valence that had a non-recursive computational form[7]. That is, they may have myopically savored information about a high-value individual prize regardless of the rewards that would accrue down the line. An alternative possibility is that participants incorrectly reasoned about uncertainty. While more studies are needed to settle this question, several aspects of our data support the latter account. First, participants were not sensitive to the valence of the information in the previous trial as may be expected from a reinforcement-learning account. Second, many participants inconsistently switched strategies across tasks, inconsistent with a purely preference-based explanation. Finally, as we discuss further below, personality metrics accounted for the sensitivity to uncertainty but not for EV. Thus, at least some of our participants may have used imperfect heuristics to prospectively reason about the information gains of alternative observations.

This view is consistent with previous studies suggesting that identifying informative observations requires difficult mental simulations of future events[22] and can be suboptimal in question-asking tasks[15,23,24]. However, while previous studies used settings with high computational complexity in which participants had to evaluate many possible queries[24,25], our results were remarkable in showing inefficient inquiries in a very simple task with only 2 possible inquiries. This suggests that people use imperfect heuristics for reasoning about uncertainty regardless of computational complexity.

A better understanding of these heuristics is thus an important question for future research. One possibility is that rather than prospecting about future outcomes and belief states as assumed by normative models, people estimate informativeness merely based on surface descriptors. Thus, in our conditions, participants may have sampled based on combinations of EV and uncertainty simply because these were the two features describing the available lotteries. This view may also explain why participants were sensitive to the precise type of post-sampling decision (Estimate versus Intervene), which was irrelevant for optimal sampling but relevant for performing the task. Thus, a critical question for future research concerns how sampling is altered (improved or impaired) by framing effects, including the relative emphasis on various task features[15,26], the specific instrumental use of the information, and the participants' intuitions and knowledge about the meaning (semantics) of the information[27,28]. A second hypothesis is that participants extended to this question-asking scenario an approach they used in explore-exploit settings in which it is often optimal to use a best-first (not uncertainty-first) strategy[29]. Distinguishing between alternative accounts of inefficient question-asking strategies will be an important goal of future research.

Our results further suggest that the participants' task understanding affected not only how they sampled but also how they adjusted their sampling while performing a task. Although participants had no access to conventional reward feedback, they showed significant improvements during the course of instrumental blocks, suggesting that they refined their information gathering based on cues from the post-sampling decisions. In the Estimate task, participants may have monitored their confidence about estimating the sum and noted that confidence was higher after sampling the high- versus low-variance lottery. In the Intervene task, participants could have noted that the amount they recuperated was higher after observing the hiVar relative to the loVar lottery. Notably, Uncertainty-only participants showed learning in both tasks while EV-sensitive participants did so primarily in the Estimate task, suggesting that people differently weight task cues depending on their understanding of the sampling strategy.

### Individual variability and personality traits

The fact that our participants requested low-informativeness observations seems consistent with the phenomenon of information avoidance, whereby people reject information about personally relevant topics if it is expected to signal a negative outcome (e.g., a financial loss or bad medical diagnosis)[3,11,30]. While this parallel must clearly be made with caution given the very different methodologies of these tasks, our results on personality metrics provide insights into the common and different processes tapped by the tasks.

Consistent with the reports of Ho and colleagues[3] and Kobayashi et al.[7], we found no credible evidence of associations between efficient sampling and standard measures of risk attitudes or demographic factors of age, sex, and educational level. However, using a broader range of personality metrics relative to that earlier study, we found reliable associations with extraversion and need for cognition, a measure correlated with fluid intelligence[31]. Importantly, these predictors were consistent across our three tasks of information demand,

supporting the conclusion of Ho et al. that personality correlates were consistent even as the level of information avoidance could change across topics (i.e., finance vs. health vs. social perception[3]). Thus, personality metrics may capture elements of information gathering that are common across distinct methodologies and tasks.

Also instructive are the differences between our findings and those of Ho and colleagues. In contrast with our results, Ho et al. found significant associations with neuroticism or openness to experience, possibly reflecting their use of information with rich semantic and personal content. On the other hand, we made a fine-grained distinction between uncertainty and value-based motives which is difficult to achieve in verbal-based tasks, and revealed that personality traits were much more predictive of uncertainty relative to reward sensitivity.

Two consistent predictors of higher uncertainty sensitivity were low extraversion and a higher need for cognition. Interestingly, although extraversion is often linked to reward sensitivity[32] we found no credible evidence that extraversion predicted reward sensitivity in our task. We speculate that, instead, extraversion captured different styles of inquiry, perhaps related to more gregarious (extraverted) versus reflective (introverted) personalities. Together with the positive association between uncertainty sensitivity and the need for cognition, this supports the idea that efficient information gathering requires accurate reasoning about uncertainty.

We also show that uncertainty sensitivity was associated with thrill-seeking and stress tolerance, two subscales of the 5DC curiosity inventory[21]. Interestingly, the relationships were complex and included a negative association with thrill-seeking and a positive association with stress tolerance. We speculate that participants who sampled more efficiently may have been more motivated to achieve an early resolution of uncertainty and sought to avoid later uncertain (risky) decisions. These participants thus combined a greater willingness to avoid risk, captured by lower thrill-seeking, with a greater ability to engage cognitive processes for resolving uncertainty, captured by higher stress tolerance. Thus, an important question for future research concerns the relation between information gathering and attitudes to uncertainty, which are known to vary across the lifespan and in psychopatholgy[10,33,34].

In sum, we show that a critical aspect of information gathering—the ability to identify and prioritize informative observations—is shaped by uncertainty, value-based and cognitive factors and is associated with personality traits with significant implications for consequential choices in natural settings.

## Methods
### Participants
All experimental procedures were approved by the Institutional Review Board of Columbia University. Participants were recruited through the online platform Amazon Mechanical Turk and completed a battery of tasks that included the task and questionnaires we describe here and eight additional tasks we will describe in different publications. The tasks were implemented in custom software (Haratki LLC) and were presented to each participant in a randomized order over several days. Informed consent was obtained before for each participant. To ensure quality data, we limited enrollment to participants who were (self-declared) adults over 18 in the United States, and who were verified as having completed more than 100 previously approved Amazon Turk tasks with an approval rate of over 80%. In addition, we limited the minimum and maximum reaction times for each trial and invited participants to perform additional tasks only after stringent tests and quality controls designed to eliminate bots. These safeguards are beyond those that produced reliable data (comparable to that from participants tested in the laboratory) in the study of Kobayashi et al.[7].

### Sample sizes
Our sample size was based on our previous study of a similar task in non-instrumental conditions. A power analysis at a $\beta$ level of 0.9, using parameter statistics from Kobayashi et al., resulted in a minimum sample size of $n = 14$ participants to detect non-zero uncertainty sensitivity. The ultimate sample size of $n = 610$ was selected based on collaboration with other projects with which this task was presented. At a $\beta$ level of 0.9, this sample size allowed for the detection of an effect size of 14% for $n = 610$, 20% for $n = 305$ (half), and 10% for $n = 1220$ (double). The 610 participants we discuss here completed the 2-lottery task and a subset ($n = 544$) completed the 5 personality questionnaires in a separate session. Demographic data (collected from 550 participants), shows that their ages ranged between 18 and 75 years old (median category, 31–35 years old), 45% were women (55% men, 0% other), and a majority completed college (58%) or a post-graduate degree (24%) with the remaining having completed only high school (17%) or a vocational school (1%).

### Tasks
Participants completed the experiment in a single testing session that lasted approximately 45–60 min and was divided into four blocks of 126 trials each. In all blocks, a trial started with the presentation of two lotteries as described in the text (Fig. 1a). In all trials, the lotteries were easily identifiable as having, respectively, high or low variance (ranges of 120 vs. 30 points between the possible values). The total EV of the lotteries was constant at 500 points, but relative EV ($\Delta$EV, defined as the EV of the high variance lottery minus the EV of the low variance lottery) was randomly drawn with uniform probability from the following set of possible values: [−110, −90, −80, −70, −60, −50, −40, −30, −20, −10, 0, 10, 20, 30, 40, 50, 60, 70, 80, 90, 110]. Keeping the sum of EV constant simplified the design of the task and allowed us to adopt a simple criterion for magnitude judgments on the Estimate task. Although this manipulation introduced an inverse correlation between the EVs of the lotteries, it did not significantly affect the independence of the draws (high or low) that the lotteries generated.

Participants were instructed that (1) one prize will be selected from each lottery, randomly and with uniform probability and (2) the total point payoff of the trial will be the sum of the prizes. Participants then chose one lottery whose precise value they wished to reveal while remaining ignorant about the value realized from the other lottery. Following the participant's choice of a lottery, the realized prize was revealed by emphasizing it in bold font and a black arrow and removing the unrealized value from the display.

The events following the reveal differed across blocks. In the Observe block, after learning the precise value of the prize about which they inquired, participants simply progressed to the next trial. In the Estimate block, participants saw a 2nd screen displaying the revealed and hidden prizes, along with a prompt to report if they believed the sum of the prizes was higher or lower than 500 points. Participants pressed an up/down arrow to indicate their estimate, and the task advanced to the next trial without feedback regarding their guesses. In the Intervene block, participants saw a second display which, in addition to the revealed and unrevealed prizes, showed a dashed line at the EV of the inspected lottery. They indicated by pressing an arrow whether they wished to keep the realized prize or change it to the EV of the lottery, after which the task proceeded to the next trial.

### General procedures, payment, and instruction
Participants started with a block of the Observe task, followed by the Estimate and Intervene tasks (in an order that was counterbalanced across participants), and finished by repeating a block of the Observe task. At the end of each block, one trial was selected randomly from those the participants played. Both prizes were revealed for that trial, and a bonus was calculated according to the block rules (equal to the sum of the prizes for the Observe block, the sum of the prizes or 0 for,

respectively, a correct or incorrect estimate on the Estimate block, and the sum of the altered prize and the prize from the unrevealed lottery on Intervene blocks). At the end of the session, the 4 bonuses were converted to US dollars (2000 points = $1) and added to the show-up fee ($1) and the total was paid to participants via Amazon Turk.

Before each block, participants received complete instructions regarding the task, which were accompanied by screenshots illustrating the prize-generating procedure and examples of representative trials, and were self-paced with unlimited opportunity to go back to previous screens. The instructions emphasized the fact that the two prizes were independently drawn and summed on each trial and comprehensively covered the meaning of the displays, distributions of lottery values, the post-reveal choices and their implications for the possible bonus, the point-to-dollar conversion and the procedure for determining and selecting the bonus. To enable measurement of reaction times (RTs), participants were instructed to hold their hands steady on the keyboard such that their left thumb accessed the space bar, and their right hand was over the numeric keypad with the index and 4th fingers over the left arrow keys to report the reveal decisions, and the middle finger accessing both the up and down keys to report the instrumental decisions. They were informed that the reaction times could be between 0.5 and 10 s, and any key press outside of that range would cause the trial to be discarded with a warning of, respectively, "Too slow" or "Too fast", and repeated at the end of the block.

After finishing the instructions, participants answered three quiz questions about each block. If they answered any question incorrectly, they received an explanation about the correct answer and attempted the question again until they gave the correct answer. After the quiz, each participant completed two practice trials (with the option to request two additional trials) and, when ready, pressed the spacebar to start the experimental block. The instructions about the information sampling stage were presented in detail before the first Observe block and repeated in the abbreviated form before each consecutive block.

**Personality questionnaires.** As part of the task battery, participants completed four personality questionnaires, namely, (1) Behavioral Inhibition System/Behavioral Approach System (BISBAS); (2) Five Dimensional Curiosity (5DC); (3) Big Five; and (4) Need for Cognition. Typically, the questionnaires were completed on a separate day either before or after the main task. Data from questionnaires were collected from a subset of $N = 501$ participants.

**Data analysis**
Data were analyzed in MATLAB version R2020a using custom code and built-in functions as specified in each case. Unless otherwise noted, all statistical comparisons used two-tailed non-parametric tests.

Modeling of choice data was based on trials that passed selection criteria (RTs between 0.5 and 10 s) and used maximum *a posteriori* estimation. Fitting was conducted by use of MATLAB's built-in mle function. We initially modeled each participant's data with a two-parameter logistic regression:

$$p(\text{reveal}) = \frac{1}{1 + \exp(-b(a + \Delta EV))} \quad (1)$$

in which $p(\text{reveal})$ is the probability of revealing the hiVar lottery, $\Delta EV$ is standardized between −1 and 1 and $a$, $b$ are free parameters estimating, respectively, the sensitivity to uncertainty and EV. A weakly regularizing, normally distributed prior was used for both $a$ and $b$ ($\mathcal{N}(0,5)$ in both cases). This facilitated model convergence in cases of near-exclusive preference without unduly influencing less extreme cases. Parameters $a$ and $b$ were deemed significant if their 95% Cis did not include 0.

The resulting parameter estimates are shown in the left column of Fig. S1a, with each point colored according to %reveal the hiVar lottery (our model-free measure of sampling efficiency). Colored stripes are approximately parallel to the ordinate, showing that the procedure separated the two factors, with efficiency being captured mainly by parameter $a$ regardless of variations in slope ($b$), as intended. However, for participants with very low slopes ($b \sim 0$), high efficiency was captured by large absolute values of $a$ that could be either positive or negative (note the points with $a < 0$). This reflected an ambiguity in the mathematical fit whereby slopes $b$ very close to zero could have positive or negative values as a function of residual noise, which then dictated the sign of parameter $a$. This sign-flipping tendency persisted even after we introduced regularizing priors.

To circumvent this ambiguity, we used an alternative method, in which we compared for each participant the fit using Eq. (1) with that from a univariate logistic regression:

$$p(\text{reveal hiVar}) = \frac{1}{1 + e^{(-c)}} \quad (2)$$

where $c$ captures a constant rate of revealing the hiVar lottery regardless of $\Delta EV$ (also computed using maximum *a posteriori* estimation and placing a prior of $\mathcal{N}(0,5)$ on $c$). Parameter $c$ was deemed significant if its 95% CI did not include 0.5 (chance). To determine which model best-captured behavior, we approximated the Bayes Information Criterion (BIC) as

$$BIC(H_i) = -2\log(L_i) + k_i \cdot \log(n) \quad (3)$$

(where, $H_i$ is the model, $L_i$ is the a posteriori likelihood, $k_i$ is the number of free parameters, and $n$ is the number of data points) and categorized a participant as EV-sensitive or EV-insensitive if the BIC favored, respectively, the bivariate or univariate model.

As shown in the right column in Fig. S1a, this alternative method largely eliminated the indeterminacy. Most of the participants who were formerly fit with negative intercepts were now categorized as EV-insensitive (triangles, plotted at $b = 0$) and were fit with parameter $c$ that was positive, and which monotonically captured sampling efficiency.

The criterion we used to define the two categories assumed that participants were EV-sensitive by default and only categorized them as EV-insensitive if there was strong evidence favoring the univariate model (BIC difference > 4.6 in favor of the latter, equivalent to Bayes factor = 10). However, the results were unchanged if we assumed instead that participants were EV-insensitive by default and only categorized them as EV-sensitive if there was strong evidence favoring the bivariate model (BIC difference < −4.6, Bayes factor = 0.1). As shown in Fig. S1b for the Estimate task, most BIC differences were far from either criterion (mean absolute difference, 91.8 ± 3.4) and 93% of participants had the same classification across the criteria, and the same was true for the Observe (94.3 ± 3.9, 88%) and Intervene task (86.2 ± 3.0, 94%).

Questionnaires were scored according to published instructions for each. The resulting 15 scores were combined with demographic indicators for age, sex, and education (see above) and used to train a support vector machine (SVM) classifier to predict the probability of sampling the quantity of interest (%reveal hiVar or %reveal hiEV defined as high or low based on a median split) using 100 bootstrap iterations with 50:50 cross-validation. We evaluated the excess accuracy for each bootstrap by randomly shuffling the parameter labels and taking the difference between the proportion of correct classifications from the unscrambled and scrambled datasets. The coefficients of each trait were stored for each bootstrap and their significance was determined by comparing 95% confidence intervals against zero. As alternative methods, we repeated the analysis using logistic classification and used linear regression to fit %reveal hiVar and

%reveal hiEV as a linear combination of the 18 predictors. The prediction accuracy as well as the sign and significance of the coefficients replicated the SVM results in all cases.

To calculate the %variance explained by each questionnaire score, we used multiple regression analysis and computed the difference in $R^2$ between a model that included all the scores and a model that included all the scores except the one of interest. We applied this method, with consistent results, to the model free-measures (%reveal hiVar and %reveal hiEV) and to the estimated parameters (combining EV-sensitive and EV-insensitive participants by pooling parameters $a$ and $c$ and assigning $b = 0$ to Uncertainty-only participants).

### Reporting summary

Further information on research design is available in the Nature Portfolio Reporting Summary linked to this article.

### Data availability

Source data are provided with this paper. The processed lottery and questionnaire data generated in this study have been deposited in the EBRAINS Database. Access can be obtained through https://doi.org/10.25493/ZQZM-PPS. The raw lottery and questionnaire data are protected and are not available due to data privacy laws.

### Code availability

Custom MATLAB scripts to preprocess data, generate psychometric fits, and conduct the support vector machine analysis have been deposited in the Code Ocean Database. Access to the code can be obtained through https://doi.org/10.24433/CO.3745345.v1.

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

### Acknowledgements

The research described in this paper was supported by a Human Frontier Research Award (RGP0018/2016), and seed grants from the Research Initiative of Science and Engineering and the Data Science Institute at Columbia University. We thank Jochen Hartman for his expert contributions to coding and implementing the task. We thank Michael

Woodford, Michael Cohanpour, Charley Wu, Hayley Jach, and Kou Murayama for valuable discussions on earlier versions of the manuscript.

## Author contributions

J.G. conceived the experiment, J.G. and L.H. designed the experiment, L.H. collected the data, I.R., G.J., and J.G. analyzed the data, I.R. drafted the manuscript, J.G. wrote the manuscript with input from all co-authors.

## Competing interests

The authors declare no competing interests.
