## [Peer Review File · Nature Communications]

Inefficient prioritization of task-relevant attributes during instrumental information demandREVIEWER COMMENTS

Reviewer #1 (Remarks to the Author):

The authors investigate different strategies underlying which information people sample information about. In previous work they had shown that when people are rewarded based on the sum of two lotteries that varied in expected value and uncertainty, participants would inefficiently sample based on value rather than uncertainty. Here, they build on this work and show that this holds also when sampling is instrumental. They compared sampling using the original non-instrumental paradigm to a task in which participants needed to estimate the mean payoff and accurately compare it to a reference to receive it, and another task, where they could swap the sampled draw against the expected value of that lottery to optimize their payoff. In both cases the optimal strategy would be to sample the uncertain lottery and yet, (even though uncertainty based sampling increased compared to the non-instrumental task) a substantial proportion of participants sampled based on expected value instead. The authors show that these strategies are predicted by personality traits, whereby higher indicators of reward sensitivity were associated with less efficient sampling and traits like need for cognition with more.

I think this is an interesting and well designed study that shows convincingly biases in information sampling and individual differences in information sampling strategies.

I don't really have much constructive criticism or suggestions to offer. I have found that any questions I had along the road were addressed through thorough additional analyses and/or in depth discussion. I believe this paper makes a valuable contribution to our understanding of how people learn about the world to guide their behavior.

Reviewer #2 (Remarks to the Author):

In this paper "Inefficient prioritization of task-relevant attributes during instrumental information demand" the authors show the following:

1. Subjects will often select information that can positively impact anticipatory utility, even though such choices in this task lead to tangible loss.
2. This tendency is related to personality traits.

The findings provide a nice addition to the large literature showing that people care about the likely valence of information when making information foraging choices. The authors should be commended for their clever and careful design. Yet, as detailed below the findings do not meet the threshold NC usually strives for in terms of "representing a sufficiently striking advance to justify publication in Nature Communications". Thus, while the study should certainly be published, it seems like it would be a better fit for a specialized journal.

In relation to the first finding – the authors claim to novelty is that "In natural settings, people decide not only when to request information, but also which attribute of a situation to inquire about. Little is known about how participants prioritize inquiries about task-relevant features." (abstract).

I appreciate that in this task subjects decide whether to receive information about one of two lotteries (prioritizing either EV or uncertainty) rather than make a binary yes/no decision about whether to receive information on lotteries in which EV and uncertainty alter parametrically (as in the many studies by Kobayashi & Hsu; van Lieshot et al., Charpentier et al.; Bromberg-Martin et al, Golman et al., and so on.). This is a nice design. But, it is not fundamentally different, in terms of the decision making process or conclusions, from studies in which subjects make yes/no decision. In both cases an option has a value and a cost that must be compared. Deciding between A and B, involves comparing the value of A to B. Whether B is another information option (as in this study) or instead a cost (such as paying for information or effort cost) is not conceptually different and the take-home is

the same – people care about EV and about uncertainty.

They show the above is true even when information is instrumental ('this persisted in instrumental conditions', abstract). But again, we know this is so from studies using lotteries (Kobayashi & Hsu, 2019) and other controlled tasks (see work by Russell Golman), and studies looking at real-life decisions such as acquiring information about HIV tests, as well as survey studies (Kelly & Sharot, 2021).

The second finding – the relationship to personality traits – is less convincing. First, a theoretical (or practical) justification for testing for this relationship is not provided. The authors mention personality traits only in the last sentence of the introduction as an after-thought. "Finally, the inefficiency was associated with personality traits related to extraversion, need for cognition and reward and uncertainty sensitivity." – they do not say why they do this. Second, the authors should replicate the findings. They have a large N, but that is not a substitute for replication. Third, it would be good to know if the relationship exists using a simple correlation, or does it only show up using a support vector machine? Forth, relating individual differences in information demand to personality traits suggests that these differences are trait-like. To make such a claim the authors should show the differences are stable across time and across task/domain. If the relationship is only seen using this exact task the importance of the finding is largely diminished. This is especially important to show as some authors (Ho, Hagmann, & Loewenstein, 2021) have claimed individual differences in information demands are not stable across domains and are thus of limited interest.

Reviewer #3 (Remarks to the Author):

This is a very interesting and valuable study that uses an elegant experimental design to probe informational preferences under closely matched non-instrumental and instrumental conditions. They show that most participants prefer to query information about the most uncertain portion of a lottery but the majority of participants also prefer to query information about high expected value portions of a lottery. Remarkably, participants as a whole only modestly altered their preferences under instrumental information demand where the former queries are optimal to maximize reward and the latter queries have no value, thus missing out on a lot of reward. This study makes further valuable contributions by finding a way to classify people based on their information preference strategies and relating these strategies to personality traits. This is an important contribution to our emerging understanding of the nexus of instrumental and non-instrumental information seeking and how valuation strategies are similar and differ in the two cases and across individuals. I also appreciate that the authors have written the paper lucidly with clear explanations with examples to make it easy to understand their hypotheses and insights.

Overall this study is definitely promising and I expect will be an influential paper when published. That said, at present there are a few aspects that could use further work to strengthen and solidify the results, interpretations, and conclusions.

Major comments

1. EV sensitivity and inefficiency. As currently written, the paper frequently mixes these concepts and treats them either equivalently or as if one necessarily implies the other (e.g. "The participants inefficiency in gathering information did not result from random or generalized exploration (which were found in only a minority of participants) but from a type of directed exploration that focused on the less informative component of the total payoff."). However, I cannot find a clear statement or proof of how strongly these are related in this data and it does not seem obvious to me this relationship is strictly necessary in this task design. How much of the inefficiency is caused by EV sensitivity? The paper needs this and would be strengthened considerably by clarifying this both at the empirical level using the data and the conceptual level for interpreting the results.

As I understand it, to maximize reward the optimal strategy is to reveal the hiVar lottery (and then use that information appropriately, which most participants did nearly all the time) and the advantage of this over the alternative action (revealing the loVar lottery) is the same regardless of the lottery EVs. Hence if two participants have the same average % reveal hiVar and the same probability of using that information appropriately, then they will have the same efficiency, regardless of whether one participant has more or less EV sensitivity than the other (slope of the sigmoid). This means it is possible for participant A to have higher EV sensitivity than participant B while also having higher efficiency due to higher average % reveal hiVar. In principle one could even imagine a participant pool with a distribution of strategies such that EV sensitivity is positively related to efficiency due to participants with high slopes tending to have high intercepts.

Of course it is true mathematically for any given participant that holding the intercept constant and adding a slope to the psychometric function will reduce the total % reveal hiVar because the psychometric function saturates (assuming a positive intercept and that delta EV are sampled symmetrically around 0, which are true in this experiment). However it is unclear to me how much of the total variation in inefficiency of participants in this task is due to variation in slopes, intercepts, or a combination of both. Looking at the data plots it seems plausible that the majority of the inefficiency in this setting could be due to low intercepts rather than high slopes.

One finding actually seems to strongly suggest this. They report that they were able to use personality data to very significantly predict high %hiVar sampling ($p < 10^{-17}$) but they found no significant prediction of EV sensitivity (p value not reported). This is very puzzling if EV sensitivity is the major cause of inefficiency in this task because % hiVar sampling is very strongly related to efficiency in this task design (perfectly related if we assume that participants use the information appropriately which most nearly always do). However this would make sense if the major cause of inefficiency in this task is low intercepts rather than high slopes.

I should acknowledge that the supplementary table does clearly show that EV sensitive participants were on average less efficient than Uncertainty only participants. However the criterion used to classify participants was asymmetric with Unc only participants having to pass strict statistical tests. Thus it is possible that this difference in efficiency is in part due to the participants in the Uncertainty only class being selected for having high uncertainty effects and as a result being highly efficient, not due to the participants in the EV sensitive class being selected for high EV effects and as a result being highly inefficient. Thus how much EV vs Uncertainty sensitivity are responsible is unclear. On this point I would find it more compelling to show a relationship between efficiency and slope and intercept parameters rather than efficiency and classes.

2. Classification and interpretation. As currently written, the paper can be taken to imply that a major finding is a discovery that participants can be classified as having one of two strategies, EV sensitive or Uncertainty Only. This classification is used as a basis to reach several conclusions including that strategies are predicted by personality traits and are consistent under non-instrumental and instrumental information demand.

However there is no qualitative or statistical evaluation of this classification scheme. Is it a good one? How well does it capture the variance across individuals within and between classes? How does it compare with alternative classifications?

If the authors want to argue that these are 'real' and hence that human behavior falls into two discrete classes then they need to provide evidence for this by evaluating the classification, for example by showing that parameters such as EV sensitivity are bimodal or at least very skewed as one would expect if there were a limited set of discrete strategies (which does not seem to be the case in Figure 4B), or that this classification performs better than alternative models of the population (e.g. in which uncertainty sensitivity and EV sensitivity each vary independently along a spectrum).

On the other hand if the authors do not mean to imply that the classes are 'real' and are simply using them as a rough way of describing and grouping participants based on their EV sensitivity, similar to a

simple grouping of participants based on the fitted slope parameter, then this is fine, but it should be stated and this should be made clear in the interpretations.

In either case it would be helpful to clarify this point by showing a simple scatterplot of EV and Uncertainty sensitivity for all participants in each block, following the example of the nice plot in Figure 2 of Kobayashi et al paper by some of the same authors which this work builds on, and which made the structure of the population visually clear.

3. Classification asymmetry. The classification is intuitive at first glance but is constructed asymmetrically which made it hard for me to interpret in some cases. The experiment has two variables of interest, EV and Uncertainty. If they were treated equally and symmetrically then there would be four classes (EV+Uncertainty, EV only, Uncertainty only, and Random) and participants would be placed in these classes by applying the same statistical thresholds to EV and Uncertainty sensitivity respectively. However, the paper treats them unequally by using only three classes (EV sensitive, Uncertainty only, and Random) and uses an asymmetric statistical requirement where participants must pass a statistical test to adjudicate between uncertainty sensitivity and random behavior but are treated as EV sensitive by default (the methods state "This procedure assumes that choices are EV-sensitive by default and only categorizes them as EV-insensitive if there is strong evidence favoring that decision").

In principle this classification could be justified (e.g. by evaluation of these classification schemes as suggested above), but at present the paper leaves it unclear how well it models the data. This caused me some issues interpreting the results including the following.

First, the classification assumes that choices are EV sensitive by default and hence effectively treats EV sensitivity as the null hypothesis which must be rejected by a strict statistical test. However, this is the opposite of the angle in the title, abstract, and motivation of the paper, which put forth the null hypothesis of an EV insensitive efficient strategy and then seek to reject that null by showing that many participants use EV sensitive inefficient strategies. From this viewpoint the classification may exaggerate the fraction of participants who are EV sensitive by treating that as the null rather than the alternative hypothesis.

Second, in theory the asymmetric statistical requirement could confound the analysis of strategy changes in Figure 4D. Suppose participants have no net tendency to change strategies over time. It seems possible that you would still see the result in Figure 4D because of the asymmetric effect of sampling noise on classification. Participants are classified as EV sens by default but are only classified as Unc only if they pass statistical tests. In addition, the two classes have different base rates (roughly 80/20 ratio of EV sens vs Unc only in Figure 3). Thus it is not surprising that most participants who are EV sens in any one given block are also EV sens in the other block (this would occur even under a null hypothesis due to the high base rate of EV sens) and that many participants who are Unc only in any one given block are EV sens in the other block (due to the difficulty of being consistently classified as Unc only except for the small subset of participants with very high Unc sensitivity). However this would be due to asymmetric classification not the temporal order of blocks.

I believe this potential confound can be controlled for by changing Figure 4D so the percentages are computed across the whole table rather than across rows to make this a symmetrical comparison between the total number of people changing from EV sens to Unc vs the opposite. Based on the numbers in the cells I expect that the left table result would no longer be significant (17 vs 20) but the right table result might become significant (11 vs 31) so hopefully the result will still hold up.

4. Terminology and interpretation. There are a few points where the wording seems a bit too strong as if presuming the psychological mechanisms for the results.

Several places use wording that seems to assume that EV sensitivity necessarily arises from anticipatory utility or savoring. These are certainly possible psychological mechanisms underlying EV

sensitivity but it would feel more precise to say the findings are consistent with them rather than proving them. For example I like the discussion paragraph about this starting on line 368, which is a nice discussion of an important insight from this study. But I would prefer the authors to insert a phrase qualifying this, e.g. "our findings imply that, if this preference is mediated by anticipatory utility, then the computational form differs substantially..."

Also is it okay to use "directed exploration" like this? In theory the term fits the experiment because participants are exploring lotteries in a directed manner. But I have primarily seen the term used in explore-exploit settings where lotteries can be chosen repeatedly and exploring implies choosing a lottery and receiving its result and learning from it to inform future choices, but not in settings like this one where lotteries are trial-unique and the action is revealing a portion of a lottery's outcome.

Minor:

line 241-242 "while the sensitivity to EV was robust in all tasks, it was slightly reduced in instrumental conditions." This wording sounds slightly off for the reasons given above in major comment 1. If taken literally I cannot find a direct statistical test of this claim so one needs to be made for this. I do agree that the percentage of participants classified as EV sensitive versus Uncertainty only decreased. However that is different from this claim that EV sensitivity was reduced. According to Figure 4 which is cited immediately after this claim, for participants who were classified as EV sensitive in all tasks EV sensitivity (slope) was not reduced by instrumental conditions relative to the Observe condition, instead it was not significantly affected (Estimate task) or slightly increased (Intervene task). The reason for improved efficiency in this population does not appear to be to reduced EV sensitivity but rather to increased Uncertainty sensitivity (intercept). It is possible that this conclusion would differ when considering the whole population but examining the average psychometric curves in Figure 3 seems potentially consistent with this conclusion as well.

paragraph starting line 284. This seems like a very valuable control but I can't quite follow the exact logic of the claim. Can this be explained in more detail? I think the idea is that in the task the hiVar lottery completely determines whether the total payoff is greater than 500 while the loVar lottery has no effect. So if you nonetheless obey the loVar lottery that suggests that you mistakenly believe the loVar draw is what determines whether the total payoff is greater than 500.

line 305 - this seems likely but the authors should present statistical evidence for this claim and report if it is significant. Visually there does seem to be a higher choice of hiVar after a hi draw than a lo draw, with small error bars suggesting that it is not implausible for there to be a small but significant effect.

line 310 - "these changes were strongest in participants with uncertainty-only strategies". This seems likely but the authors should present statistical evidence for this claim, since there is a significant change in the EV sensitive group too in at least one block.

line 325 - "This above-chance accuracy could not be explained by a spurious correlation between uncertainty and EV sensitivity, as we found no significant prediction of EV sensitivity." If true this is an important, striking, and surprising result suggesting that personality traits may be more linked to hiVar sensitivity than EV sensitivity, despite the presence of traits that a priori ought to help such as reward responsiveness. The paper would be strengthened by showing the null result for EV sensitivity in the same format as the positive result for hiVar sensitivity for instance in the same format as Figure 5A and reporting significance of a test comparing the two results.

line 358-367 - "A majority of participants queried the less informative prize if it had higher EV. People who pursued less efficient strategies had higher levels of the trait BAS reward, and higher scores on the trait extraversion that is associated with reward sensitivity¹⁴, supporting the view that the inefficiency was explained by sensitivity to anticipatory utility. The EV sensitivity was significant in all tasks". This is the issue from major comment 1. The phrasing seems to equate "inefficiency" with "EV

sensitivity" and then uses this to claim that the personality traits identified by the SVM are predictive of the EV sensitivity. but the results section claims that the SVM only predicted % reveal hiVar (a close proxy for efficiency) and was unable to predict EV sensitivity.

lines 404-410 It is surprising that according to their interpretation participants with a higher index of uncertainty tolerance were more likely to approach uncertain options in order to learn about them and achieve an early resolution of uncertainty. Naively one would expect people with low uncertainty tolerance to have the strongest preference to resolve uncertainty. This would be worth discussing.

line 425 I am not sure what "sizeable fraction" of participants they are referring to who were "having less efficient strategies in instrumental relative to non-instrumental tasks". The plot I see quantifying something most like this is 4C but it only shows 7 participants who used an uncertainty only strategy in the non-instrumental task but not in the instrumental tasks. Also if my earlier comments are correct the EV sensitivity does not necessarily imply inefficiency of strategy as long as % reveal hiVar remains constant or increases so participants who became EV sensitive were not necessarily becoming less efficient. My suggestion is to come up with a numerical measure of efficiency of strategy. Then they can do a straightforward and direct test of their claim by testing how many individuals were more efficient in instrumental vs non-instrumental tasks.

line 592 - "tolerance intervals". Are these really tolerance intervals and not just standard bootstrap confidence intervals? I was expecting confidence intervals since they are what I have usually seen in similar analyses and they seem appropriate since they are asking if the parameter of interest is different from 0. If using tolerance intervals the authors should explain why they are appropriate here and specify the tolerance interval fully (a tolerance interval is defined by two parameters, a proportion of the population and the confidence level with which the interval contains it). If using confidence intervals can they clarify how the p values were computed? The methods says significance was determined by 90% intervals but the figure says it shows 95% intervals and $p < 0.05$.

line 795 - "salience" is used here and in methods but not elsewhere. I guess this wording may be left over from an earlier version of the manuscript before terminology was changed.

Reviewer #1

The authors investigate different strategies underlying which information people sample information about. In previous work they had shown that when people are rewarded based on the sum of two lotteries that varied in expected value and uncertainty, participants would inefficiently sample based on value rather than uncertainty. Here, they build on this work and show that this holds also when sampling is instrumental. They compared sampling using the original non-instrumental paradigm to a task in which participants needed to estimate the mean payoff and accurately compare it to a reference to receive it, and another task, where they could swap the sampled draw against the expected value of that lottery to optimize their payoff. In both cases the optimal strategy would be to sample the uncertain lottery and yet, (even though uncertainty based sampling increased compared to the non-instrumental task) a substantial proportion of participants sampled based on expected value instead. The authors show that these strategies are predicted by personality traits, whereby higher indicators of reward sensitivity were associated with less efficient sampling and traits like need for cognition with more.

I think this is an interesting and well designed study that shows convincingly biases in information sampling and individual differences in information sampling strategies.

I don't really have much constructive criticism or suggestions to offer. I have found that any questions I had along the road were addressed through thorough additional analyses and/or in depth discussion. I believe this paper makes a valuable contribution to our understanding of how people learn about the world to guide their behavior.

We thank the reviewer for these supportive comments.

Reviewer #2:

In this paper "Inefficient prioritization of task-relevant attributes during instrumental information demand" the authors show the following:

1. Subjects will often select information that can positively impact anticipatory utility, even though such choices in this task lead to tangible loss.
2. This tendency is related to personality traits.

The findings provide a nice addition to the large literature showing that people care about the likely valence of information when making information foraging choices. The authors should be commended for their clever and careful design. Yet, as detailed below the findings do not meet the threshold NC usually strives for in terms of "representing a sufficiently striking advance to justify publication in Nature Communications". Thus, while the study should certainly be published, it seems like it would be a better fit for a specialized journal.

We thank the reviewer for these candid comments. The perception of novelty is of course, subjective and we believe that the results may be more innovative than the reviewer suggests. Nevertheless, the comment suggests that we did not make this sufficiently clear, and we extensively revised the manuscript to convey what we see as the most innovative features.

In relation to the first finding – the authors claim to novelty is that “In natural settings, people decide not only when to request information, but also which attribute of a situation to inquire about. Little is known about how participants prioritize inquiries about task-relevant features.” (abstract). I appreciate that in this task subjects decide whether to receive information about one of two lotteries (prioritizing either EV or uncertainty) rather than make a binary yes/no decision about whether to receive information on lotteries in which EV and uncertainty alter parametrically (as in the many studies by Kobayashi & Hsu; van Lieshot et al., Charpentier et al.; Bromberg-Martin et al, Golman et al., and so on.). This is a nice design. But, it is not fundamentally different, in terms of the decision making process or conclusions, from studies in which subjects make yes/no decision. In both cases an option has a value and a cost that must be compared. Deciding between A and B, involves comparing the value of A to B. Whether B is another information option (as in this study) or instead a cost (such as paying for information or effort cost) is not conceptually different and the take-home is the same – people care about EV and about uncertainty.

We agree with the reviewer that all decisions including those in our tasks are characterized by tradeoffs between value and costs. However, we respectfully suggest that this may be an overly broad criterion by which to judge novelty. We can think of no study of decision making, including those that appear in *Science* and *Nature* (or are described in the *New York Times!*), that cannot be interpreted in terms of tradeoffs between value and costs. The core questions are about **the precise ways in which people assess values and costs**, which are extremely diverse and can be very difficult to define.

In our specific experiment, the novel aspect of estimating information value is the need to compare and integrate the informativeness two attributes that pertain to one situation (two lotteries that add to one payoff). This process is not captured by traditional studies of information demand that focus strictly on one-attribute situations. The latter studies use accept/reject decisions as the reviewer states, but their fundamental difference with our experiment is that they only require judgment about one informative feature (lottery). Provided that the participant obtains information about that one feature, they can perfectly resolve the uncertainty of the situation.

This is a big simplification relative to natural tasks in which participants judge complex situations and, because of limitations of time and resources, can only partly resolve the uncertainty about the situation. This poses a need for information selection: deciding which attribute to attend to, by estimating how the uncertainty of individual attributes contributes to the uncertainty of the full situation.

The vast majority of normative models of choice, including reinforcement learning models and economic models, make the assumption that participants achieve this integration in a recursive fashion. That is, the models assign value to an informational state by combining all the values that are expected to accrue at downstream stages that follow this state. This is a core assumption that ensures computational tractability and inter-temporal choice consistency.

As we showed in Kobayashi et al. for non-instrumental conditions, our results imply that this is too restrictive an assumption for describing human information gathering. Instead, humans myopically demand information about individual observations, without properly inferring the implications of this information about the full situation. Perhaps the most concrete expression of this difference is the fact that the model of anticipatory utility of ligaya et al., which predicts value-based biases in tasks with individual lotteries, does *not* predict the value biases in our task because of its normative (recursive) assumption.

Thus, our present results have an important novel implication. They show that, even in instrumental conditions that are properly incentivized, human information gathering violates a major assumption of

normative decision models and reflect non-recursive valuation of informational states. Although this is an empirical rather than computational modeling paper, we believe that the findings will motivate significant changes in future computational models of information gathering.

In the revised manuscript we endeavored to better emphasize this point by expanding the discussion about information integration in the *Introduction* (pages 2-4) and additionally emphasizing the fact that this integration requires a cognitive prospection about future states that is cognitively effortful in both the *Results* (**Fig. 1B** and **Fig. 3C,D** and related text), and throughout the *Discussion*.

They show the above is true even when information is instrumental ('this persisted in instrumental conditions', abstract). But again, we know this is so from studies using lotteries (Kobayashi & Hsu, 2019) and other controlled tasks (see work by Russell Golman), and studies looking at real-life decisions such as acquiring information about HIV tests, as well as survey studies (Kelly & Sharot, 2021).

We agree with the reviewer that the studies cited above examined the demand for instrumental information. However, we are not aware that they examined the integration of information across attributes. Kobayashi and Hsu 2019 presented individual lotteries as in traditional studies. As far as we know, the same is true for work from by Rus Golman and colleagues, although we would be happy to consider any specific study the reviewer may know of that we missed. Finally, Kelly and Sharot 2021 used surveys in which information integration across attributes was indeed likely to occur but could not be quantitatively characterized in terms of value/reduction of uncertainty as we did in our task.

The second finding – the relationship to personality traits – is less convincing. First, a theoretical (or practical) justification for testing for this relationship is not provided. The authors mention personality traits only in the last sentence of the introduction as an after-thought. "Finally, the inefficiency was associated with personality traits related to extraversion, need for cognition and reward and uncertainty sensitivity." – they do not say why they do this.

This is a good point and we thank the reviewer for bringing it up. In response, we greatly expanded our justification of measuring personality traits in the *Introduction*, where we point out how these questionnaires can link studies of information demand that use very different methodologies (i.e., simulated naturalistic scenarios vs pared-down laboratory tasks). In addition, we added an entirely new section to the *Discussion* in which we compare in detail our findings with those of Ho and colleagues. We hope these revisions will enhance the interest in our paper in a broader range of scholars studying information demand with a wider range of methodologies.

Second, the authors should replicate the findings. They have a large N, but that is not a substitute for replication.

We believe we provided convincing replication by testing 3 different tasks – including a non-instrumental and two instrumental conditions. These tasks were of course similar in terms of information demand as needed for a controlled study, but they were quite different in terms of the subsequent decisions and elicited significant differences in results. Thus, we believe that the tasks demonstrate reliable replication. Please note that this strategy goes beyond most papers that use a single task and, we believe, is comparable to that of Ho and colleagues who replicated the findings by testing information demand across 3 different topics (information about personal finance, health or social perceptions).

In the revised manuscript, we emphasize this point as applicable in the *Abstract*, *Introduction*, *Results* and at the start of the *Discussion*.

Third, it would be good to know if the relationship exists using a simple correlation, or does it only show up using a support vector machine?

We had found that the associations with personality scores replicated in two additional methods – logistic classification and multiple regression analyses. We now include this information in the relevant parts of the *Results* and *Methods* sections.

We also noted that Ho et al. in their 2021 study (see below) argued that personality scores show low correlations with task measures, indicating that these are not exchangeable constructs. We thought this was an excellent point, and conducted an additional regression analysis computing the variance explained by each personality score. This is now described in the last paragraph in the *Methods* and the first paragraph in the *Results* section on personality metrics. This corroborated the conclusion of Ho et al. by showing that no personality score explained more than 2.5% of the variance in uncertainty and reward sensitivity in our tasks.

Forth, relating individual differences in information demand to personality traits suggests that these differences are trait-like. To make such a claim the authors should show the differences are stable across time and across task/domain. If the relationship is only seen using this exact task the importance of the finding is largely diminished. This is especially important to show as some authors (Ho, Hagmann, & Loewenstein, 2021) have claimed individual differences in information demands are not stable across domains and are thus of limited interest

We thank the authors for alerting us to the 2021 study of Ho, Hagmann and Loewenstein, of which we were not aware. We believe the study is quite relevant to our findings and we based our revision strongly on it, including a section in which we discuss in detail the relation between its findings and our results.

It is our understanding that Ho et al. showed that information demand varied across topics (finance, health and social perceptions) but the association with personality scores remained constant. As we now point out in the *Discussion*, we obtained similar results, finding that information demand could vary across the 3 tasks (Observe, Estimate and Intervene) but the associations with personality traits remained fairly consistent.

As we now discuss at length, we found a partial but not perfect correspondence between the personality indicators that we and Ho et al. found to be predictive. However, we believe that both the similarities and differences are very instructive and can provide valuable bridges between investigations of information demand that have used very different methods and have remained so far separate. In the revised *Discussion*, the reviewer can find our detailed arguments on this point.

Reviewer #3 (Remarks to the Author):

This is a very interesting and valuable study that uses an elegant experimental design to probe informational preferences under closely matched non-instrumental and instrumental conditions. They show that most participants prefer to query information about the most uncertain portion of a lottery but the majority of participants also prefer to query information about high expected value portions of a lottery. Remarkably, participants as a whole only modestly altered their preferences under instrumental information demand where the former queries are optimal to maximize reward and the latter queries have no value, thus missing out on a lot of reward. This study makes further valuable contributions by

finding a way to classify people based on their information preference strategies and relating these strategies to personality traits. This is an important contribution to our emerging understanding of the nexus of instrumental and non-instrumental information seeking and how valuation strategies are similar and differ in the two cases and across individuals. I also appreciate that the authors have written the paper lucidly with clear explanations with examples to make it easy to understand their hypotheses and insights.

Overall this study is definitely promising and I expect will be an influential paper when published. That said, at present there are a few aspects that could use further work to strengthen and solidify the results, interpretations, and conclusions.

We thank the reviewer for their supportive comments and constructive criticisms listed below. We endeavored to address these criticisms in detail. We believe the revisions strengthened the paper and hope that they addressed the reviewer's concerns.

1. EV sensitivity and inefficiency. As currently written, the paper frequently mixes these concepts and treats them either equivalently or as if one necessarily implies the other (e.g. "The participants inefficiency in gathering information did not result from random or generalized exploration (which were found in only a minority of participants) but from a type of directed exploration that focused on the less informative component of the total payoff."). However, I cannot find a clear statement or proof of how strongly these are related in this data and it does not seem obvious to me this relationship is strictly necessary in this task design. How much of the inefficiency is caused by EV sensitivity? The paper needs this and would be strengthened considerably by clarifying this both at the empirical level using the data and the conceptual level for interpreting the results. As I understand it, to maximize reward the optimal strategy is to reveal the hiVar lottery (and then use that information appropriately, which most participants did nearly all the time) and the advantage of this over the alternative action (revealing the loVar lottery) is the same regardless of the lottery EVs. Hence if two participants have the same average % reveal hiVar and the same probability of using that information appropriately, then they will have the same efficiency, regardless of whether one participant has more or less EV sensitivity than the other (slope of the sigmoid). This means it is possible for participant A to have higher EV sensitivity than participant B while also having higher efficiency due to higher average % reveal hiVar. In principle one could even imagine a participant pool with a distribution of strategies such that EV sensitivity is positively related to efficiency due to participants with high slopes tending to have high intercepts.

Of course it is true mathematically for any given participant that holding the intercept constant and adding a slope to the psychometric function will reduce the total % reveal hiVar because the psychometric function saturates (assuming a positive intercept and that delta EV are sampled symmetrically around 0, which are true in this experiment). However it is unclear to me how much of the total variation in inefficiency of participants in this task is due to variation in slopes, intercepts, or a combination of both. Looking at the data plots it seems plausible that the majority of the inefficiency in this setting could be due to low intercepts rather than high slopes. One finding actually seems to strongly suggest this. They report that they were able to use personality data to very significantly predict high %hiVar sampling ($p < 10^{-17}$) but they found no significant prediction of EV sensitivity (p value not reported). This is very puzzling if EV sensitivity is the major cause of inefficiency in this task because % hiVar sampling is very strongly related to efficiency in this task design (perfectly related if we assume that participants use the information appropriately which most nearly always do). However

this would make sense if the major cause of inefficiency in this task is low intercepts rather than high slopes.

I should acknowledge that the supplementary table does clearly show that EV sensitive participants were on average less efficient than Uncertainty only participants. However the criterion used to classify participants was asymmetric with Unc only participants having to pass strict statistical tests. Thus it is possible that this difference in efficiency is in part due to the participants in the Uncertainty only class being selected for having high uncertainty effects and as a result being highly efficient, not due to the participants in the EV sensitive class being selected for high EV effects and as a result being highly inefficient. Thus how much EV vs Uncertainty sensitivity are responsible is unclear. On this point I would find it more compelling to show a relationship between efficiency and slope and intercept parameters rather than efficiency and classes.

We fully agree with the reviewer that we had not properly defined what we mean by “sampling efficiency”. We also agree that, although efficiency and EV sensitivity are distinct constructs as the reviewer states, our writing throughout had confused them.

We rectified these concerns by changing our wording everywhere to clearly refer to either uncertainty sensitivity/efficiency or EV sensitivity. We explicitly mention that we define “efficiency” using a model-free measure of % revealing hiVar (the frequency of revealing the high variance lottery) and justify our choice (*Results*, p. 9, 2nd paragraph). In addition, we added statements making clear that reward and uncertainty sensitivity are distinct constructs, throughout the *Introduction*, *Results* and *Discussion*. We hope that these changes address this very valid concern.

2. Classification and interpretation. As currently written, the paper can be taken to imply that a major finding is a discovery that participants can be classified as having one of two strategies, EV sensitive or Uncertainty Only. This classification is used as a basis to reach several conclusions including that strategies are predicted by personality traits and are consistent under non-instrumental and instrumental information demand. However there is no qualitative or statistical evaluation of this classification scheme. Is it a good one? How well does it capture the variance across individuals within and between classes? How does it compare with alternative classifications? If the authors want to argue that these are 'real' and hence that human behavior falls into two discrete classes then they need to provide evidence for this by evaluating the classification, for example by showing that parameters such as EV sensitivity are bimodal or at least very skewed as one would expect if there were a limited set of discrete strategies (which does not seem to be the case in Figure 4B fig 4B is relative to observe), or that this classification performs better than alternative models of the population (e.g. in which uncertainty sensitivity and EV sensitivity each vary independently along a spectrum). On the other hand if the authors do not mean to imply that the classes are 'real' and are simply using them as a rough way of describing and grouping participants based on their EV sensitivity, similar to a simple grouping of participants based on the fitted slope parameter, then this is fine, but it should be stated and this should be made clear in the interpretations. In either case it would be helpful to clarify this point by showing a simple scatterplot of EV and Uncertainty sensitivity for all participants in each block, following the example of the nice plot in Figure 2 of Kobayashi et al paper by some of the same authors which this work builds on, and which made the structure of the population visually clear.

We agree with the reviewer that the original draft had an incomplete justification of our rationale for and our interpretation of the classification method we used. In the revision, we greatly expand on this point. We added a new supplementary figure (**Fig. S2**) showing the detailed plots the reviewer requests, of

parameters a and b for each separate task and fitting method. We also added a new panel in the main text (new **Fig. 2B**) explaining how the fitting procedure we chose maps into the model-free measure of efficiency (%reveal hiVar). Finally, we added a several paragraphs in the *Discussion* (p. 15/16) clarifying our interpretation of the results.

As is now outlined in these sections, our method was motivated by a technicality associated with the fitting procedure (which was otherwise similar to that used in Kobayashi et al.) Specifically, we found that, when fit by bi-variate sigmoid functions, a sizeable minority of participants with weak sensitivity to EV produced slopes that were near zero but could take positive or negative values. This in turn forced the (much larger) intercepts to be, correspondingly, positive or negative, and created a non-monotonic relationship between the intercepts and %reveal hiVar, whereby high efficiency could correspond to positive or negative intercepts. While we had encountered this phenomenon in the data of Kobayashi et al., here it became more pronounced because of our use of instrumental conditions that enhanced uncertainty sensitivity, and it could no longer be satisfactorily resolved using regularization parameters as had been the case in the original study.

To mitigate this difficulty, we turned to the current model-selection procedure, where we fit each participant with the better of two fits –the bivariate sigmoid or a univariate function with only a (positive) intercept, c . As we show in new **Fig. S2**, this method reduced the indeterminacy in the bi-variate fits and captured %reveal hiVar monotonically through the intercepts (a or c ; compare panels **Fig. S2A** and **S2B**).

This method naturally lent itself to classifying participants as EV-sensitive or EV-insensitive according to their best-fit function, and we believe that this classification captured meaningful aspects of our data. Briefly, participants in the “Random” category who were insensitive to either uncertainty or EV showed other behavioral markers consistent with being inattentive or disengaged from the task (i.e., low accuracy for their post-sampling decisions (**Fig. S1**) & no significant learning during a block; **Fig. S7B**). Second, participants classified as “Uncertainty-only” had %reveal hiVar values greater than 90% and formed a distinct mode of the overall distribution of model-free sampling efficiency (**Fig. 2B**). Finally, the vast majority of participants classified as EV-sensitive showed a range of reward and uncertainty sensitivity consistent with the findings of Kobayashi et al. The full arguments for this view are now laid out detail in the *Results* related to **Fig. 2B** and at the start of the *Discussion*.

In sum, the classification method was motivated by a technical aspect of the fitting procedure, and, we believe, captured meaningful aspects of behavior within the constraints of our task.

3. Classification asymmetry. The classification is intuitive at first glance but is constructed asymmetrically which made it hard for me to interpret in some cases. The experiment has two variables of interest, EV and Uncertainty. If they were treated equally and symmetrically then there would be four classes (EV+Uncertainty, EV only, Uncertainty only, and Random) and participants would be placed in these classes by applying the same statistical thresholds to EV and Uncertainty sensitivity respectively. However, the paper treats them unequally by using only three classes (EV sensitive, Uncertainty only, and Random) and uses an asymmetric statistical requirement where participants must pass a statistical test to adjudicate between uncertainty sensitivity and random behavior but are treated as EV sensitive by default (the methods state "This procedure assumes that choices are EV-sensitive by default and only categorizes them as EV-insensitive if there is strong evidence favoring that decision"). In principle this classification could be justified (e.g. by evaluation of these classification schemes as suggested above), but at present the paper leaves it unclear how well it models the data. This caused me some issues interpreting the results including the following.

First, the classification assumes that choices are EV sensitive by default and hence effectively treats EV sensitivity as the null hypothesis which must be rejected by a strict statistical test. However, this is the opposite of the angle in the title, abstract, and motivation of the paper, which put forth the null hypothesis of an EV insensitive efficient strategy and then seek to reject that null by showing that many participants use EV sensitive inefficient strategies. From this viewpoint the classification may exaggerate the fraction of participants who are EV sensitive by treating that as the null rather than the alternative hypothesis.

We agree we should have better justified our choice of classification criterion, and we added this justification in **Fig. S2C**. Specifically, we compared two criteria that assumed that choices are EV-sensitive or, alternatively, EV-*ins*sensitive by default and requiring substantive model evidence to switch the classification. We show that, in each task, over 90% of participants fall unambiguously into the same category under either criterion. Thus, our choice of the former criterion did not substantially alter our conclusions or results.

Second, in theory the asymmetric statistical requirement could confound the analysis of strategy changes in Figure 4D. Suppose participants have no net tendency to change strategies over time. It seems possible that you would still see the result in Figure 4D because of the asymmetric effect of sampling noise on classification. Participants are classified as EV sens by default but are only classified as Unc only if they pass statistical tests. In addition, the two classes have different base rates (roughly 80/20 ratio of EV sens vs Unc only in Figure 3). Thus it is not surprising that most participants who are EV sens in any one given block are also EV sens in the other block (this would occur even under a null hypothesis due to the high base rate of EV sens) and that many participants who are Unc only in any one given block are EV sens in the other block (due to the difficulty of being consistently classified as Unc only except for the small subset of participants with very high Unc sensitivity). However this would be due to asymmetric classification not the temporal order of blocks.

I believe this potential confound can be controlled for by changing Figure 4D so the percentages are computed across the whole table rather than across rows to make this a symmetrical comparison between the total number of people changing from EV sens to Unc vs the opposite. Based on the numbers in the cells I expect that the left table result would no longer be significant (17 vs 20) but the right table result might become significant (11 vs 31) so hopefully the result will still hold up.

We believe that this comment may reflect a misunderstanding of our initial presentation of **Fig. 4D**. The normalization procedure we adopted did control for the differences in base rates of EV-sensitive versus Uncertainty-only strategies.

To make this clearer, we simplified the table to show the fraction of participants who repeated a strategy in the 2nd block, *conditioned* on having pursued an EV-sensitive or Uncertainty-only strategy in the first block. We hope that this makes it easier to see that, by conditioning on the initial strategy we removed the base-rate effect.

As described in the accompanying text, we find that repeating an Uncertainty-only strategy was no more likely than repeating an EV-sensitive strategy. Our normalization procedure rules out the possibility that this is an artefact of base rates. Instead, it shows that those (few) participants who started with an Uncertainty-only strategy were no more likely to repeat this strategy than the (more

numerous) participants who started with an EV-only strategy were to repeat their strategy. We hope that the revisions to the graphics and text for **Fig. 4D** now clearly convey this point.

4. Terminology and interpretation. There are a few points where the wording seems a bit too strong as if presuming the psychological mechanisms for the results.

Several places use wording that seems to assume that EV sensitivity necessarily arises from anticipatory utility or savoring. These are certainly possible psychological mechanisms underlying EV sensitivity but it would feel more precise to say the findings are consistent with them rather than proving them. For example I like the discussion paragraph about this starting on line 368, which is a nice discussion of an important insight from this study. But I would prefer the authors to insert a phrase qualifying this, e.g. "our findings imply that, if this preference is mediated by anticipatory utility, then the computational form differs substantially..."

We agree and softened our language throughout to refer to our findings as "reward sensitivity", "value sensitivity" or "EV sensitivity" rather than savoring or anticipatory utility. In addition, we make it more clear in the *Introduction* and *Discussion* that our phenomenon does not appear equivalent to anticipatory utility.

Also is it okay to use "directed exploration" like this? In theory the term fits the experiment because participants are exploring lotteries in a directed manner. But I have primarily seen the term used in explore-exploit settings where lotteries can be chosen repeatedly and exploring implies choosing a lottery and receiving its result and learning from it to inform future choices, but not in settings like this one where lotteries are trial-unique and the action is revealing a portion of a lottery's outcome.

We agree. We moved the mention of these constructs into the new section on personality metrics and added a few words explaining where we see the connections (*Discussion*, p 20, penultimate paragraph).

Minor:

line 241-242 "while the sensitivity to EV was robust in all tasks, it was slightly reduced in instrumental conditions." This wording sounds slightly off for the reasons given above in major comment 1. If taken literally I cannot find a direct statistical test of this claim so one needs to be made for this. I do agree that the percentage of participants classified as EV sensitive versus Uncertainty only decreased. However that is different from this claim that EV sensitivity was reduced. According to Figure 4 which is cited immediately after this claim, for participants who were classified as EV sensitive in all tasks EV sensitivity (slope) was not reduced by instrumental conditions relative to the Observe condition, instead it was not significantly affected (Estimate task) or slightly increased (Intervene task). The reason for improved efficiency in this population does not appear to be to reduced EV sensitivity but rather to increased Uncertainty sensitivity (intercept). It is possible that this conclusion would differ when considering the whole population but examining the average psychometric curves in Figure 3 seems potentially consistent with this conclusion as well.

Consistent with our response to comment 1, we changed the language throughout to carefully distinguish between EV sensitivity and efficiency (which is equivalent to uncertainty sensitivity).

paragraph starting line 284. This seems like a very valuable control but I can't quite follow the exact logic of the claim. Can this be explained in more detail? I think the idea is that in the task the hiVar lottery completely determines whether the total payoff is greater than 500 while the loVar lottery has no effect. So if you nonetheless obey the loVar lottery that suggests that you mistakenly believe the loVar draw is what determines whether the total payoff is greater than 500.

We also think this is a valuable control, and have revised the text to explain it more clearly (*Results*, top of p. 12). The idea is that people may believe that an observation from the *loVar* lottery reliably predicts the total payoff (because it predicts the *hiVar* observation). However, if that were the case, we expect that, in the Estimate task, participants should show high confidence in their estimates of the sum after a *loVar* observation. Our analyses suggest this is not the case, since reaction times for estimating the sum are longer after a *loVar* relative to a *hiVar* observation, and participants who choose *loVar* observations more frequently are *less* likely to estimate sums congruently with these observations (i.e., to guess that the total is high when the revealed prize is high and vice versa). In other words, participants seem to obtain *loVar* observations even though they have less confidence in their estimation decisions than after obtaining a *hiVar* observation, arguing against the idea that they believe that a *loVar* observation is predictive of the *hiVar* prize or of the total payoff.

line 305 - this seems likely but the authors should present statistical evidence for this claim and report if it is significant. Visually there does seem to be a higher choice of hiVar after a hi draw than a lo draw, with small error bars suggesting that it is not implausible for there to be a small but significant effect.

We believe this comment reflects a slight misreading of old **Fig. S6A** (new **Fig. S7A**). The red and green bars in the figure indicate the participants' sampling decision on the previous trial, and the two bars within each color indicate the prize that was observed (high versus low) given that observation.

The red bars are slightly higher than the green ones simply because of consistency (autocorrelation) in the choices (i.e., if a person selected the hiVar lottery in one trial they are slightly more likely to select it again in the next trial than they are to switch to the loVar lottery). The key difference, however, is between observing a low or high prize given each choice (the two red bars and between the two green bars). This factor clearly has no effect or an interaction with the choice.

To avoid this misunderstanding, we added labels making it clear that the previous trial was categorized according to the choice (red vs green) and the observation obtained after making the choice (two bars within each color).

line 310 - "these changes were strongest in participants with uncertainty-only strategies". This seems likely but the authors should present statistical evidence for this claim, since there is a significant change in the EV sensitive group too in at least one block. reword

We present the statistical comparisons of learning effects at the group and individual levels in the legend to **Fig. S8** and direct the reader to this figure in the *Results*.

line 325 - "This above-chance accuracy could not be explained by a spurious correlation between

uncertainty and EV sensitivity, as we found no significant prediction of EV sensitivity." If true this is an important, striking, and surprising result suggesting that personality traits may be more linked to hiVar sensitivity than EV sensitivity, despite the presence of traits that a priori ought to help such as reward responsiveness. The paper would be strengthened by showing the null result for EV sensitivity in the same format as the positive result for hiVar sensitivity for instance in the same format as Figure 5A and reporting significance of a test comparing the two results. Add a/b prediction as supplementary?

This is a good point, we thank the reviewer for bringing it up. We now include the requested information as a new panel in **Fig. 4A**, expanded our description of the SVM prediction of EV sensitivity in the related parts of the *Results* and add a paragraph on this point in the *Discussion* (p. 18/19).

line 358-367 - "A majority of participants queried the less informative prize if it had higher EV. People who pursued less efficient strategies had higher levels of the trait BAS reward, and higher scores on the trait extraversion that is associated with reward sensitivity¹⁴, supporting the view that the inefficiency was explained by sensitivity to anticipatory utility. The EV sensitivity was significant in all tasks". This is the issue from major comment 1. The phrasing seems to equate "inefficiency" with "EV sensitivity" and then uses this to claim that the personality traits identified by the SVM are predictive of the EV sensitivity. but the results section claims that the SVM only predicted % reveal hiVar (a close proxy for efficiency) and was unable to predict EV sensitivity. reword

Again, we agree and corrected our phrasing to clearly distinguish sampling efficiency (%reveal hiVar) from EV sensitivity (which we measure here with %reveal higherEV and also with parameter *b*).

lines 404-410 It is surprising that according to their interpretation participants with a higher index of uncertainty tolerance were more likely to approach uncertain options in order to learn about them and achieve an early resolution of uncertainty. Naively one would expect people with low uncertainty tolerance to have the strongest preference to resolve uncertainty. This would be worth discussing.

We had offered a discussion of this point in the previous version – and we now moved it to the *Discussion* (p. 19). As the reviewer can see, we propose that the intuition the reviewer mentions is captured by the construct of thrill seeking. We speculate that people with low thrill seeking aim to resolve uncertainty early and, provided they also have high stress tolerance, are willing to approach the *hiVar* lottery in order to do so by gathering information.

line 425 I am not sure what "sizeable fraction" of participants they are referring to who were "having less efficient strategies in instrumental relative to non-instrumental tasks". The plot I see quantifying something most like this is 4C but it only shows 7 participants who used an uncertainty only strategy in the non-instrumental task but not in the instrumental tasks. Also if my earlier comments are correct the EV sensitivity does not necessarily imply inefficiency of strategy as long as % reveal hiVar remains constant or increases so participants who became EV sensitive were not necessarily becoming less efficient. My suggestion is to come up with a numerical measure of efficiency of strategy. Then they can do a straightforward and direct test of their claim by testing how many individuals were more efficient in instrumental vs non-instrumental tasks.

The reviewer is correct that this was not good wording on our part and we removed this argument from the text (having also justified our measure of “efficiency” as we noted above).

line 592 - "tolerance intervals". Are these really tolerance intervals and not just standard bootstrap confidence intervals? I was expecting confidence intervals since they are what I have usually seen in similar analyses and they seem appropriate since they are asking if the parameter of interest is different from 0. If using tolerance intervals the authors should explain why they are appropriate here and specify the tolerance interval fully (a tolerance interval is defined by two parameters, a proportion of the population and the confidence level with which the interval contains it). If using confidence intervals can they clarify how the p values were computed? The methods says significance was determined by 90% intervals but the figure says it shows 95% intervals and $p < 0.05$.

We thank the reviewer for spotting this. The reviewer is correct that we used 95% confidence intervals everywhere, and we have now corrected the erroneous statements in the *Methods*.

line 795 - "salience" is used here and in methods but not elsewhere. I guess this wording may be left over from an earlier version of the manuscript before terminology was changed.

Thanks for the good catch! The term “salience” was left over from a previous draft and we hope we finally removed all the references to it in the current version.

REVIEWER COMMENTS

Reviewer #2 (Remarks to the Author):

The authors addressed my technical concerns.

To be honest, my overall assessment had not changed, but of course that is subjective.

The one thing the authors did not address is insufficient engagement with the existing literature. All the papers I had cited in the previous review are relevant and should be cited. There are many other papers as well that have shown the importance of uncertainty and anticipatory affect (or EV) on information seeking, as well as papers that show how these factors are combined to direct information-seeking. In the second paragraph, for example, in which EV is first mentioned no citation is given of EV directing information-seeking.

Reviewer #3 (Remarks to the Author):

The authors have extensively revised their manuscript with very careful and thorough attention to my comments and suggestions. Great job! I especially appreciate the new phrasing more clearly distinguishing inefficiency from EV sensitivity, the nice plots and justifications for their model fitting and classification procedure in Figure 2B and Figure S2, and the new plot in Figure 4A and associated descriptions.

I have mostly minor comments and suggestions.

1. The authors addressed my first major comment very well overall. However, one aspect was not addressed. What portion of the inefficiency was due to intercept (lower than 100% reveal of hiVar when relative value = 0) vs slope (additional reduction in % reveal of hiVar when considering cases where relative value \neq 0)?

In my view this is an important question to ask in this study. The study's main topic is the inefficiency (as stated in the title), and the main analysis is based on models with two parameters (intercept and slope) that together control the inefficiency. Since the authors find clear effects of both parameters and they vary across tasks, it seems important to give readers at least a rough idea of what balance of the two parameters are responsible for the inefficiency and how this changed between the non-instrumental vs instrumental tasks. For example, my impression is that the improvement in efficiency going from non-instrumental to instrumental tasks is more due to higher intercepts (more Uncertainty sensitivity) than lower slopes (less EV sensitivity). Whatever the answer, I would like it if this was reported for readers to see.

Here is my suggestion for a hopefully simple and easy way to report a rough measure of this. For each of the average psychometric curves in 2A, calculate the overall % reveal hiVar (I believe this is simply the average of all the data points of the curve) as an estimate of their total efficiency, and then calculate the % reveal hiVar using only the data from the curve at $x=0$ as an estimate of what their total efficiency would have been if all participants had ignored EV differences between lotteries (and hence as if there was no influence of EV sensitivity). Then report those two numbers, and what fraction the former is of the latter. (Of course this could be more accurate if using curve-fitting to estimate the intercepts instead of using the raw data values at $x=0$)

2. "An alternative possibility is that participants improved their sampling simply by spending more time on the task and reflecting on the optimal strategy...Similarly, we found no evidence that efficiency improved after participants experienced the instrumental conditions – i.e., in the second relative to the first Observe blocks (Fig. S8)."

I am not sure this plot provides clear evidence on this. The plot is trying to compare behavior in the first vs last Observe blocks. However the curves being compared in each plot have different n so they

are not the same sets of participants. I assume this is because the plots only compare participants within classes and classification is done separately for each participant in each block. Hence the results are not conclusive since participants may not have improved efficiency by changing their sensitivities within class but rather by moving between classes.

My read of the plot seems consistent with that alternate interpretation. Comparing the N the Observe Repeat block has fewer participants classified as EV sensitive and more classified as Uncertainty Only. This seems potentially consistent with participants "spending more time on the task and reflecting on the optimal strategy" to shift away from EV sensitivity toward uncertainty sensitivity.

My suggestion to make this analysis more clear and convincing is to simply compare the grand average psychophysical curves in the first vs last Observe blocks including all participants, and directly test whether the total efficiency (total % reveal hiVar) over all participants was significantly different between the two blocks.

As a minor note I am confused why the EV sensitive curve in panel A has $N = 411$ but the table in panel C says 339/347. I would have expected it to be something/411.

3. In the discussion of the ligaya et al model, I am not sure the reasoning in lines 423-428 is valid for why the model would predict that sampling is insensitive to relative EV. The reasoning is based on the two lotteries being equally likely to signal high or low prizes or positive or negative RPEs. It is true that that model uses the (absolute value of) the RPEs to boost the value of anticipation, but the value of anticipation for a reward also scales with the size of that reward (page 3 of his paper). So even if two situations have equal 50/50 chances of positive vs negative RPEs I would still expect their model to predict higher anticipatory value for the situation with higher expected reward.

The reasoning in lines 429-435 is more convincing to me to justify this conclusion. I agree that the ligaya model's anticipatory value should be based on the total reward from the current state (pooling both lotteries) and hence may not be related to the difference in reward between the two lotteries (the relative EV parameter studied in this paper).

4. The authors cite and thoroughly discuss a study by Ho et al linking personality traits to information search. It would be good to cite other recent work that has also related personality traits to information search. Two that come to mind are

Bennet et al J Exp Psychol Gen 2021

Kelly and Sharot, Nat Commun 2021

Very minor comments:

Lines 246-247 have missing words or a typo.

Line 434 per my previous comments, savoring is one of several possible psychological mechanisms, so I would replace "our results show" with something like "our results suggest" or "if savoring is responsible, our results show".

Figure S2A scatterplots are missing labels on the x and y axes.

Reviewer #2:

The authors addressed my technical concerns.

To be honest, my overall assessment had not changed, but of course that is subjective.

The one thing the authors did not address is insufficient engagement with the existing literature. All the papers I had cited in the previous review are relevant and should be cited. There are many other papers as well that have shown the importance of uncertainty and anticipatory affect (or EV) on information seeking, as well as papers that show how these factors are combined to direct information-seeking. In the second paragraph, for example, in which EV is first mentioned no citation is given of EV directing information-seeking.

In the previous version, we omitted some individual citations (of which there are many indeed) in favor of several broad reviews that prominently discuss them. At the reviewer's request, we now added the citations to Kobayashi & Hsu, 2019, Kelly & Sharot, 2021 and work by Russell Golman. These citations are highlighted in red in the reference list and referenced at the appropriate places in the text.

Reviewer #3 (Remarks to the Author):

The authors have extensively revised their manuscript with very careful and thorough attention to my comments and suggestions. Great job! I especially appreciate the new phrasing more clearly distinguishing inefficiency from EV sensitivity, the nice plots and justifications for their model fitting and classification procedure in Figure 2B and Figure S2, and the new plot in Figure 4A and associated descriptions.

Thank you for these supportive comments.

I have mostly minor comments and suggestions.

1. The authors addressed my first major comment very well overall. However, one aspect was not addressed. What portion of the inefficiency was due to intercept (lower than 100% reveal of hiVar when relative value = 0) vs slope (additional reduction in % reveal of hiVar when considering cases where relative value \sim 0)? In my view this is an important question to ask in this study. The study's main topic is the inefficiency (as stated in the title), and the main analysis is based on models with two parameters (intercept and slope) that together control the inefficiency. Since the authors find clear effects of both parameters and they vary across tasks, it seems important to give readers at least a rough idea of what balance of the two parameters are responsible for the inefficiency and how this changed between the non-instrumental vs instrumental tasks. For example, my impression is that the improvement in efficiency going from non-instrumental to instrumental tasks is more due to higher intercepts (more Uncertainty sensitivity) than lower slopes (less EV sensitivity). Whatever the answer, I would like it if this was reported for readers to see. Here is my suggestion for a hopefully simple and easy way to report a rough measure of this. For each of the average psychometric curves in 2A, calculate the overall % reveal hiVar (I believe this is simply the average of all the data points of the curve) as an estimate of their total efficiency, and then calculate the % reveal hiVar using only the data from the curve at $x=0$ as an estimate of what their total efficiency would have been if all participants had ignored EV differences between lotteries (and hence as if there was no influence of EV sensitivity). Then report those two numbers, and what fraction the former is of the latter. (Of course this could be more accurate if using curve-fitting to estimate the intercepts instead of using the raw data values at $x=0$)

We performed precisely the analysis the reviewer suggests and present the results on p. 9 (bottom). We thank the reviewer for this idea, and agree that the results will be interesting to many readers.

2. "An alternative possibility is that participants improved their sampling simply by spending more time on the task and reflecting on the optimal strategy...Similarly, we found no evidence that efficiency improved after participants experienced the instrumental conditions – i.e., in the second relative to the first Observe blocks (**Fig. S8**)."

I am not sure this plot provides clear evidence on this. The plot is trying to compare behavior in the first vs last Observe blocks. However the curves being compared in each plot have different n so they are not the same sets of participants. I assume this is because the plots only compare participants within classes and classification is done separately for each participant in each block. Hence the results are not conclusive since participants may not have improved efficiency by changing their sensitivities within class but rather by moving between classes. My read of the plot seems consistent with that alternate interpretation. Comparing the N the Observe Repeat block has fewer participants classified as EV sensitive and more classified as Uncertainty Only. This seems potentially consistent with participants "spending more time on the task and reflecting on the optimal strategy" to shift away from EV sensitivity toward uncertainty sensitivity. My suggestion to make this analysis more clear and convincing is to simply compare the grand average psychophysical curves in the first vs last Observe blocks including all participants, and directly test whether the total efficiency (total % reveal hiVar) over all participants was significantly different between the two blocks.

This is a good point. We analyzed the combined data and included a note on p. 33 that we found no difference in overall %hiVar sampling from the first to the last Observe blocks.

As a minor note I am confused why the EV sensitive curve in panel A has N = 411 but the table in panel C says 339/347. I would have expected it to be something/411.

We apologize for this oversight. Panel C indeed had an error, which we now corrected (and added a reference to it in the text below, highlighted at the bottom of p. 33). Thank you for the good catch!

3. In the discussion of the ligaya et al model, I am not sure the reasoning in lines 423-428 is valid for why the model would predict that sampling is insensitive to relative EV. The reasoning is based on the two lotteries being equally likely to signal high or low prizes or positive or negative RPEs. It is true that that model uses the (absolute value of) the RPEs to boost the value of anticipation, but the value of anticipation for a reward also scales with the size of that reward (page 3 of his paper). So even if two situations have equal 50/50 chances of positive vs negative RPEs I would still expect their model to predict higher anticipatory value for the situation with higher expected reward. The reasoning in lines 429-435 is more convincing to me to justify this conclusion. I agree that the ligaya model's anticipatory value should be based on the total reward from the current state (pooling both lotteries) and hence may not be related to the difference in reward between the two lotteries (the relative EV parameter studied in this paper).

Thank you for helping us clarify our thoughts on this point. We agree, and corrected the relevant discussion paragraph accordingly (p. 14/15). We endeavored to clarify the difference between our results and those of ligaya et al. and clearly state alternative hypotheses that are not based on anticipatory utility.

4. The authors cite and thoroughly discuss a study by Ho et al linking personality traits to information search. It would be good to cite other recent work that has also related personality traits to information search. Two that come to mind are Bennet et al J Exp Psychol Gen 2021 and Kelly and Sharot, Nat Commun 2021

We agree and have now included both references at the appropriate points in the discussion (see highlights in the reference list).

Very minor comments:

Lines 246-247 have missing words or a typo. Thank you, we added the missing words (now on line 212).

Line 434 per my previous comments, savoring is one of several possible psychological mechanisms, so I would replace "our results show" with something like "our results suggest" or "if savoring is responsible, our results show".

Thank you for pressing us on this important point. We endeavored to remove any language like the one the reviewer highlights. Moreover, we went a step further and shored up our explanation in the *Introduction*. The most critical paragraph is highlighted on p. 3 and is better prefaced by the revised preceding paragraphs. Finally, we highlight a paragraph to the *Discussion* (p. 15, starting at line 381), which had been there before but is now revised for clarity, which outlines two hypotheses – based on a non-recursive form of anticipatory utility or based on heuristic reasoning about informativeness.

We hope that these changes convey the key point that the value biases we find may differ substantially from the previously described phenomena of anticipatory utility.

Figure S2A scatterplots are missing labels on the x and y axes.

We added the labels.

Reviewers' Comments:

Reviewer #3:

Remarks to the Author:

Great work! The authors have thoroughly addressed all of my comments. This is a very nice paper that I am sure will be influential in the field.